# Associability-modulated loss learning is increased in posttraumatic stress disorder

**Vanessa M Brown[1,2], Lusha Zhu[1,3], John M Wang[1,2], B Christopher Frueh[4], Brooks King-Casas[1,2,5,6†]\*, Pearl H Chiu[1,2,5†]\***

[1]Virginia Tech Carilion Research Institute, Roanoke, United States; [2]Department of Psychology, Virginia Tech, Blacksburg, United States; [3]School of Psychological and Cognitive Sciences, IDG/McGovern Institute for Brain Research, Beijing Key Laboratory of Behavior and Mental Health, Peking–Tsinghua Center for Life Sciences, Peking University, Beijing, China; [4]University of Hawaii, Hilo, United States; [5]Salem Veterans Affairs Medical Center, Salem, United States; [6]School of Biomedical Engineering and Sciences, Virginia Tech-Wake Forest University, Blacksburg, United States

**Abstract** Disproportionate reactions to unexpected stimuli in the environment are a cardinal symptom of posttraumatic stress disorder (PTSD). Here, we test whether these heightened responses are associated with disruptions in distinct components of reinforcement learning. Specifically, using functional neuroimaging, a loss-learning task, and a computational model-based approach, we assessed the mechanistic hypothesis that overreactions to stimuli in PTSD arise from anomalous gating of attention during learning (i.e., associability). Behavioral choices of combat-deployed veterans with and without PTSD were fit to a reinforcement learning model, generating trial-by-trial prediction errors (signaling unexpected outcomes) and associability values (signaling attention allocation to the unexpected outcomes). Neural substrates of associability value and behavioral parameter estimates of associability updating, but not prediction error, increased with PTSD during loss learning. Moreover, the interaction of PTSD severity with neural markers of associability value predicted behavioral choices. These results indicate that increased attention-based learning may underlie aspects of PTSD and suggest potential neuromechanistic treatment targets.

DOI: https://doi.org/10.7554/eLife.30150.001

\*For correspondence:
bkcasas@vtc.vt.edu (BK-C);
chiup@vtc.vt.edu (PHC)

†These authors contributed equally to this work

Competing interests: The authors declare that no competing interests exist.

## Introduction

Posttraumatic stress disorder (PTSD) is debilitating and characterized by excessive behavioral, psychological, and physiological responses to unexpected stimuli (*Pitman et al., 2012*). In particular, clinical and empirical observations have documented the negative impact of salient cues on neural and behavioral functioning in PTSD, including heightened orienting to unexpected events, impaired extinction of learned fear, and unstable attention biases toward perceived threatening stimuli (*Aupperle et al., 2012*; *Bar-Haim et al., 2007*; *Blair et al., 2013*; *Morey et al., 2009*; *Naim et al., 2015*). Together, these behavioral alterations in response to unexpected stimuli, uncontrollable reminders of trauma, and other negative, threatening, and trauma-related events point to PTSD as a disorder of disrupted learning from reminders of negative events; however, the specific components of anomalous learning in PTSD remain unknown. As an initial step toward addressing this issue, we adopt a computational psychiatry approach (*Montague et al., 2012*; *Wang and Krystal, 2014*; *Maia and Frank, 2011*), using quantitative specification of neural and behavioral learning processes to investigate the neurocomputational substrates of PTSD.

**eLife digest** Posttraumatic stress disorder, or PTSD for short, is a serious psychiatric disorder that sometimes occurs after someone has experienced a dangerous or threatening event. People with PTSD are prone to overreact to unexpected reminders of these events, and are often hypervigilant for danger. Why these symptoms occur is not yet clear, but it is thought that people with PTSD may have learning problems that lead them to overestimate the likelihood of danger.

Advanced tools from computer science and mathematics have helped scientists to study how the brain learns. These tools may now provide more insight into how diseases like PTSD disrupt learning. Scientists use computer models of learning to test how humans make choices and react to their outcomes. These models build on the idea that humans make choices based on what they predict an outcome will be, and then learn when they update their expectations based on the accuracy of their predictions.

Now, Brown et al. show that people with PTSD have an increased learning response to surprising events – these are defined in this study as outcomes that are inconsistent with participants' predictions. In the experiments, 74 combat veterans who had experienced trauma in Iraq or Afghanistan underwent a type of brain scanning procedure, while they played a gambling-like game. Some participants had PTSD, others did not.

Both groups learned to make choices that minimized the loss of money. However, learning in veterans with PTSD was strongly influenced by how much attention they paid to surprising outcomes. Moreover, the brain areas that help to process attention to surprise were highly active in people with PTSD. Brown et al. added a third group of participants with depression to the study to verify that the learning changes were PTSD-specific. This depression-only group did not have differences in attention to surprise.

Many treatments for PTSD focus on exposing individuals to feared situations and trauma memories, so that individuals can learn that these situations are no longer dangerous. Computational modeling and neuroimaging may help scientists pinpoint the sources of learning deficits, such as increased attention to surprising outcomes. Identifying the different possible causes of learning problems may lead to new or more precise learning-based treatments for PTSD and other learning-related conditions. Understanding how learning-related brain changes develop may also help find ways to prevent and better diagnose PTSD and other psychiatric disorders.

DOI: https://doi.org/10.7554/eLife.30150.002

Computational model-based approaches to learning provide a mechanistic framework for understanding the detrimental impact of unexpected negative stimuli and reminders of negative events in PTSD. Error-guided models of reinforcement learning (RL) have robustly shown that unexpected outcomes (i.e., value 'prediction errors') drive learning by directly updating the value of the cues associated with those outcomes (*Rescorla and Wagner, 1972*; *Sutton and Barto, 1998*). A related family of hybrid reinforcement learning models combines prediction-error based learning with a dynamically changing attention modulation variable (i.e., a cue's associability value) that scales with the magnitude of prediction errors previously associated with a particular cue. In these models, trial-by-trial associability values associated with particular cues gate the learning of subsequent outcomes associated with these cues (*Li et al., 2011*; *Pearce and Hall, 1980*; *Le Pelley, 2004*). Thus, these models contain separate parameters for error-based learning rate and associability updating that together govern how strongly current and past prediction errors, respectively, affect learning. While commonly used tasks that assess cue-salience (*Todd et al., 2015*) and attention to threat (*Naim et al., 2015*; *Vythilingam et al., 2007*) test important components of aversive processing in PTSD, they are more limited for explaining how changes in attention to negative stimuli may affect subsequent behavioral choices, as reinforcement learning algorithms allow in the context of value-based learning tasks. More generally, computational model-based approaches allow fitting of models of neural function to behavioral choices and imaging data and thus facilitate the separation of mechanistic processes (e.g., responses to associability value, associability updating, learning rate, prediction error responses) related to attention and learning from negative and positive events. In the hybrid RL framework, PTSD, and symptoms of hypervigilance in particular, may reflect

disproportionate attentional processes that drive maladaptive, heightened responses to stimuli with a history of unexpected outcomes.

The neural substrates of reinforcement learning suggest further compelling links between associability-modulated learning and PTSD, as the brain networks involved in both overlap. In particular, work in humans and rodents has identified roles for the ventral striatum, anterior cingulate, and amygdala in encoding prediction error (*Rangel et al., 2008*; *Pagnoni et al., 2002*; *Garrison et al., 2013*) and for the amygdala and insula in encoding associability values (*Li et al., 2011*; *Roesch et al., 2012*). In PTSD, affective stimuli consistently elicit altered neural activation in a network that prominently also includes the amygdala, insula, and prefrontal regions (*Hayes et al., 2012*; *Etkin and Wager, 2007*). Of translational relevance, attention, learning, and the amygdala have all been behavioral and neural targets of promising new therapies for PTSD (*Badura-Brack et al., 2015*; *Craske et al., 2014*; *Langevin et al., 2016*); elucidating the component neurobehavioral mechanisms associated with learning in PTSD may refine these targets.

In brief, extant data indicate that neural and behavioral distinctions between solely prediction error-based and associability-modulated learning contribute uniquely to neural and behavioral correlates of learning and may clarify processes underlying the heightened sensitivity to unexpected stimuli in PTSD. To assess this possibility, we implemented a probabilistic learning task during functional magnetic resonance imaging (fMRI) in combat-deployed military veterans with and without posttraumatic stress disorder. We posited that if attention-modulated learning plays a role in PTSD, hybrid RL models incorporating associability should predict participants' choices better than learning models without associability. Furthermore, PTSD severity, particularly symptoms related to hyperarousal (*Lissek and van Meurs, 2015*), should correlate preferentially both with enhanced associability updating and with increased activity in neural structures encoding associability values (i.e., amygdala and insula). Critically, these relationships would not be expected between PTSD and solely error-based learning rate or error-related activity in neural structures encoding value and prediction error (i.e., ventral striatum, ventromedial prefrontal cortex).

## Results

### Participant characteristics and model-agnostic behavioral performance

Combat-deployed military veterans (N = 74) completed a probabilistic learning task in the loss and gain domains while undergoing fMRI scanning (*Figure 1*; full task description in Materials and methods). All veterans had served at least one tour in Iraq or Afghanistan since 2001, had experienced Criterion A deployment-related trauma, and were recruited from a larger study by our group examining biomarkers of mood and anxiety disorders. To be considered in the present analyses, participants were further required to demonstrate behavioral engagement on the relevant portions of the probabilistic learning task (N = 68 veterans; see Materials and methods for exclusion breakdowns, full inclusion/exclusion criteria, and exclusion details for sub-analyses). Veterans were assessed for PTSD using the Clinician Administered PTSD Scale (CAPS [*Blake et al., 1995*]) and for other psychiatric disorders with the Structured Clinical Interview for DSM-IV (SCID [*First et al., 1996*]). Participants exhibited a range of PTSD symptoms, with 39 veterans meeting DSM-IV (*American Psychiatric Association, 2000*) criteria for PTSD (see *Supplementary file 1*, Table 1A for clinical and demographic information); the other 29 veterans had previous deployment-related trauma exposure as assessed by the CAPS interview but did not meet criteria for PTSD, resulting in 39 participants with PTSD and 29 participants without PTSD for primary analyses.

In both loss and gain domains, veterans showed robust learning, as evidenced by increasing likelihood of choosing the 'better' option from chance on the first trial to near 80% correct after ~15 trials (*Figure 2a* and *Figure 2—figure supplement 1*). An adaptive design titrated the task for participants to achieve sufficient learning (i.e., block length was adjusted once participants achieved learning; see Materials and methods for details); reflecting this, performance accuracy (% better choice) did not differ between participants with and without PTSD in the gain or loss domains (gain: $t_{41} = -1.29$, p>0.1; loss: $t_{41} = 0.459$, p>0.1).

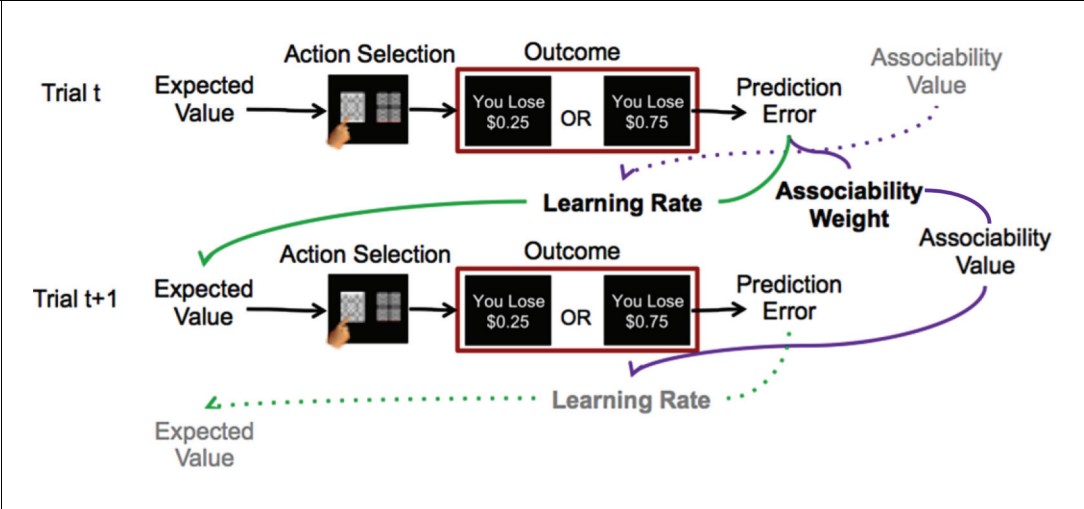

**Figure 1.** Reinforcement learning model. Schematic description of reinforcement learning model. Participants choose between two stimuli, view the monetary outcome of the choice, and learn over time which is the 'better' option. A reinforcement learning model incorporating associability-modulated learning rate is illustrated here. Expected value is updated based on the static learning rate, the current trial's prediction error, and the associability value from the previous outcome associated with the stimulus. Meanwhile, associability value is updated based on the static associability weight and the absolute value of the current trial's prediction error. Green lines indicate the effect of prediction error through learning rate on expected value, while purple lines indicate the effect of prediction error through associability weight on associability value on the current trial, and then on learning rate on the subsequent trial. Thus, through trial-by-trial modulation of learning rates, associability values act as an attentional gate on learning. The task involved blocks consisting of all loss or all gain trials; loss trials are shown here as the model with associability fit well in this condition only.

DOI: https://doi.org/10.7554/eLife.30150.003

## Behavioral model-fitting and relationship of model parameters with PTSD

As an initial step toward evaluating the role of associability in learning in PTSD, participants' choices were fit to a prediction error-based reinforcement learning (RL) model with and without a dynamic associability value ($\kappa$)-modulated learning rate on the prediction error ($\delta$), as in *Li et al. (2011)*. In this model, the associability value $\kappa$ of a chosen stimulus changes on a trial-by-trial basis based on a combination of the magnitude of previous prediction errors and a static associability weight $\eta$, a parameter which varies by participant and indicates the extent to which the magnitude of recent prediction errors updates trial-by-trial associability values (see Materials and methods for full model specifications and *Figure 2b* for a trial-by-trial plot of the time course and relationship between prediction error and associability values). We verified via simulation that associability weight ($\eta$) did not directly affect performance (*Figure 3—figure supplement 1a*), allowing us to dissociate the effects of associability updating from general performance deficits. Consistent with our hypothesis, including associability in the RL model significantly improved model fit for the majority of participants and did so during loss learning only (*Figure 3a*; protected exceedance probability of model with versus without associability: 0% in gain, 100% in loss). The role of associability in loss, but not gain, learning is consistent with prior data showing heightened orienting and attentional biases toward negative information in PTSD (*Li et al., 2011*; *Boll et al., 2013*); thus, our subsequent analyses focused on learning variables in the loss domain (see Appendix 2, Supplementary Results for supplemental data related to gain learning and model-agnostic support for the presence of associability-modulated updating during loss learning only).

The role of associability in loss learning was further corroborated by robust effects of associability values on behavior during the task. First, choices predicted by the associability RL model showed high correspondence with participants' actual choices (correlation of bins of predicted vs. observed choices: $r = 0.997$, $p<0.001$; *Figure 3—figure supplement 1b*) and did not differ between participants with and without PTSD (t-test of subjects' log likelihoods and PTSD diagnosis: $t_{66} = -0.33$, $p>0.1$). Reaction times were significantly and positively correlated with trial-by-trial associability

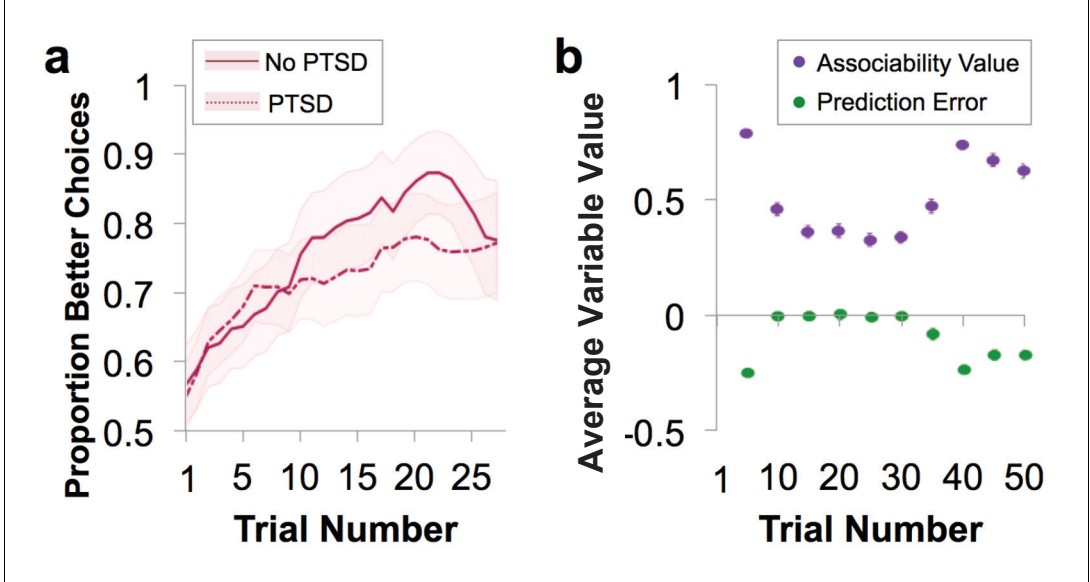

**Figure 2.** Behavioral performance and relationship to parameter estimates. (**a**) Loss performance. Performance was quantified as proportion of choices that were the 'better' option. Over time, participants show learning (running average over five trials, averaged over all blocks; mean ± SE). Behavior is separated by diagnostic group, with veteran control (No PTSD) participants' behavior marked by a solid line and the behavior of veterans with PTSD marked by a dotted line. (**b**) Plot of trial-by-trial associability value and prediction error for loss learning blocks, averaged across sets of five trials and across participants. Values are derived from the associability reinforcement learning model described in *Figure 1* using each participant's individually estimated parameter values and behavioral choices. As expected, a gradual reduction in average associability value and reduction of prediction error towards zero across trials is observed. As the initial expected value of each stimulus was set at 0, prediction errors are initially negative. During learning, prediction errors become distributed around, and converge toward, 0, indicating learning. Similarly, the initial associability value is set at one and decreases as outcomes become better predicted. Note that the probabilistic outcomes of this task ensure that outcomes are not completely predicted; this feature maintains variation in prediction error and retains the influence of associability value. As the task progresses, new blocks requiring new learning are more likely to occur, leading to an overall increase in associability value and prediction errors further from 0. Gain learning blocks did not fit the associability model well; figures related to gain learning are shown in *Figure 2—figure supplement 1*.

DOI: https://doi.org/10.7554/eLife.30150.004

The following figure supplement is available for figure 2:

**Figure supplement 1.** Gain learning performance.
DOI: https://doi.org/10.7554/eLife.30150.005

values ($\kappa$), reflecting greater decision latency as the associability value of the chosen option increased (*Figure 3b*; average per-subject regression beta value of reaction time predicting associability, controlling for expected value of chosen option and trial number: .232; t-test assessing difference from 0: $t_{68} = 8.28$, p<0.001). Next, we regressed trial-by-trial estimates of prediction error ($\delta$) and associability-modulated prediction error ($\kappa*\delta$), respectively, computed from participants' individually estimated parameter values, against their trial-by-trial switching behavior (switch or no switch, a measure of responsivity to outcomes). Associability-modulated prediction error significantly predicted switching choices above prediction error alone ($\chi^2_1 = 323.0$, p<0.001; *Figure 3c*). In addition, associability weight and other model parameters were recoverable through simulation (*Figure 3—figure supplement 1c*), and associability value showed a robust neural effect independent of PTSD diagnosis (*Figure 3d*; *Supplementary file 1*, Table 1B). These data together indicate that the associability RL model fit participants' behavior and neural activity well and support the incorporation of associability in the RL model of loss learning.

If associability-modulated learning plays a role in PTSD, individuals with PTSD ought to have greater associability weights ($\eta$; see Materials and methods for model specifications). To test this possibility, participants' individually estimated associability weights, reflecting the degree to which associability values are updated based on recent unsigned prediction errors during loss learning for each participant, and unmodulated learning rates ($\alpha$) were compared between participants with and without a PTSD diagnosis (see Appendix 1: Supplementary Methods for individual parameter

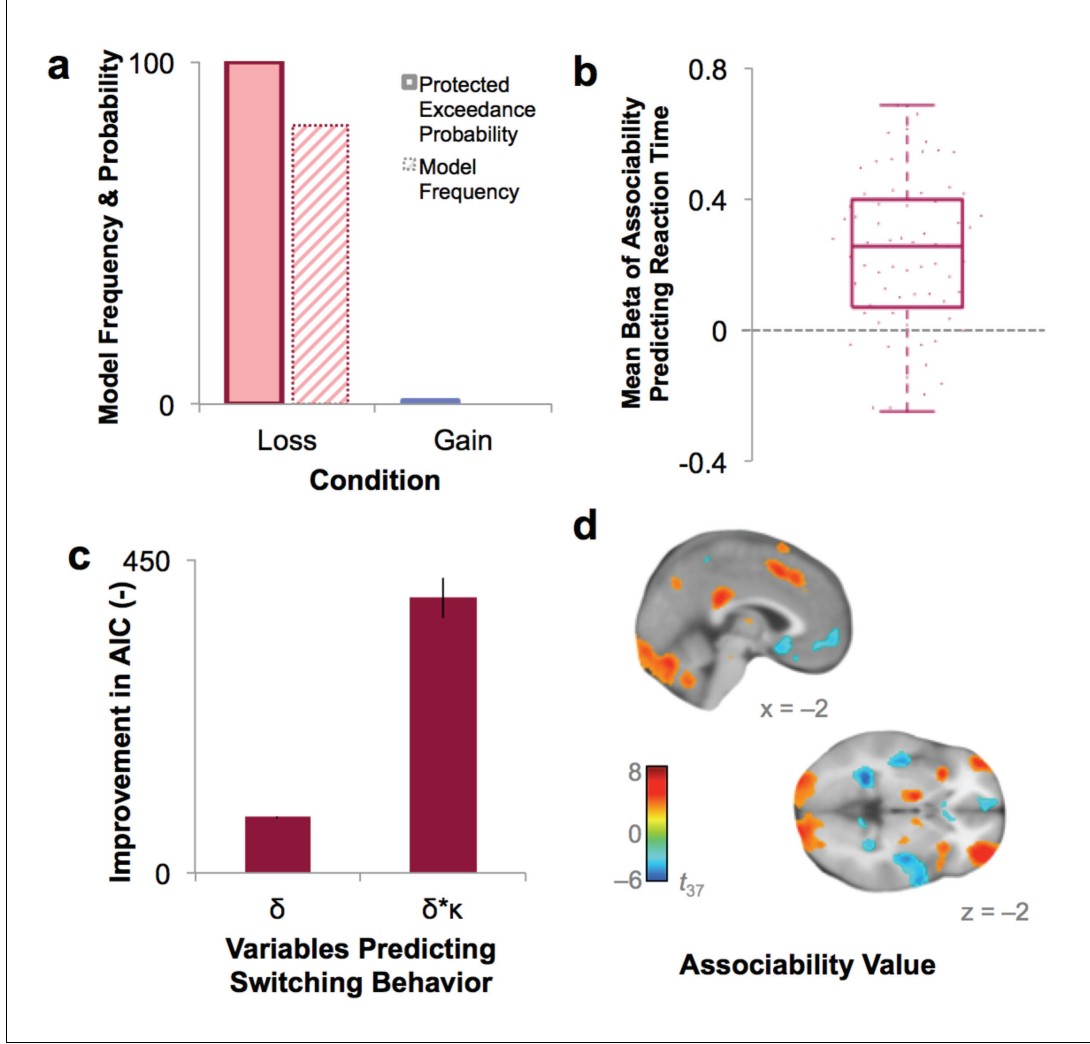

**Figure 3.** Model fit and relationship to behavior. (a) Protected exceedance probability and model frequency calculated by Bayesian Model Selection for the model with versus without associability for loss and gain trials, showing an improvement in model fit when adding associability during loss learning only. (b) Average regression beta values per subject between trial-by-trial associability value ($\kappa$) and reaction time (controlling for expected value and trial number) show a positive relationship between model-estimated associability value and choice latencies. Dots are individual subjects' beta values. (c) Prediction of switching behavior is improved when previous trial's prediction error ($\delta$) is modulated by associability value ($\kappa$; negative change in AIC for logistic regression predicting switching; $\delta$ and $\delta * \kappa$ model, each compared against basic outcome-only model; see **Supplementary** Materials and methods). Error bars represent subject-level standard errors based on a leave-one-out standard error estimation. (d) Associability signaling independent of PTSD and covariates, displayed at $p<0.05$ FDR corrected. First level regression of trial-by-trial associability values on neural activity at time of outcome. Second level (shown) of constant term of regression. Additional model confirmation analyses (relationship between associability weight parameter and performance, relationship between actual and predicted choices, and model parameter recovery) are in *Figure 3—figure supplement 1*.

DOI: https://doi.org/10.7554/eLife.30150.006

The following figure supplement is available for figure 3:

**Figure supplement 1.** Model confirmation analyses.

DOI: https://doi.org/10.7554/eLife.30150.007

estimation details). Associability weights were increased in participants with PTSD ($t_{62}$ = 4.01, p<0.001) while unmodulated learning rate did not differ between groups ($t_{62}$ = 0.63, p>0.1; *Figure 4a*), supporting the hypothesis of a greater emphasis on modulation of loss learning by attention in PTSD.

Given the high co-occurrence of depression in PTSD (*O'Donnell et al., 2004*), we also tested the specificity of increased associability weight to PTSD versus depression. Specifically, we enrolled a separate cohort of gender-, estimated IQ-, and smoking status-matched participants with a current diagnosis of major depressive disorder (MDD; N = 20; see Materials and methods for MDD participant details), but not PTSD. These MDD-only participants performed the same learning task, and we also fit the behavior of these participants to the RL model with associability. Compared to MDD-only participants, the participants with PTSD had significantly higher associability weights ($t_{57}$ = 7.25, p<0.001) but similar unmodulated learning rates ($t_{57}$ = 1.44, p>0.1). Therefore, despite the high comorbidity between PTSD and depression (*O'Donnell et al., 2004*), increased associability-modulated learning appears specific to PTSD and not to mood-related psychopathology.

## Neural substrates of associative learning in PTSD and relationship to behavioral choices

To investigate the instantiation of neural substrates of associability- and prediction error- based learning in PTSD, we first regressed trial-by-trial estimates of associability value (κ) and prediction error (δ), respectively, during loss learning against subjects' neural responses to the outcome event (per [*Li et al., 2011*; *Esber et al., 2012*; *Roesch et al., 2010a*]; see Materials and methods for design matrix specifications). Corroborating the behavioral findings of greater associability value updating in participants with PTSD, neural encoding of associability values showed a significant relationship with PTSD at an FDR-corrected whole brain significance level in a network of regions

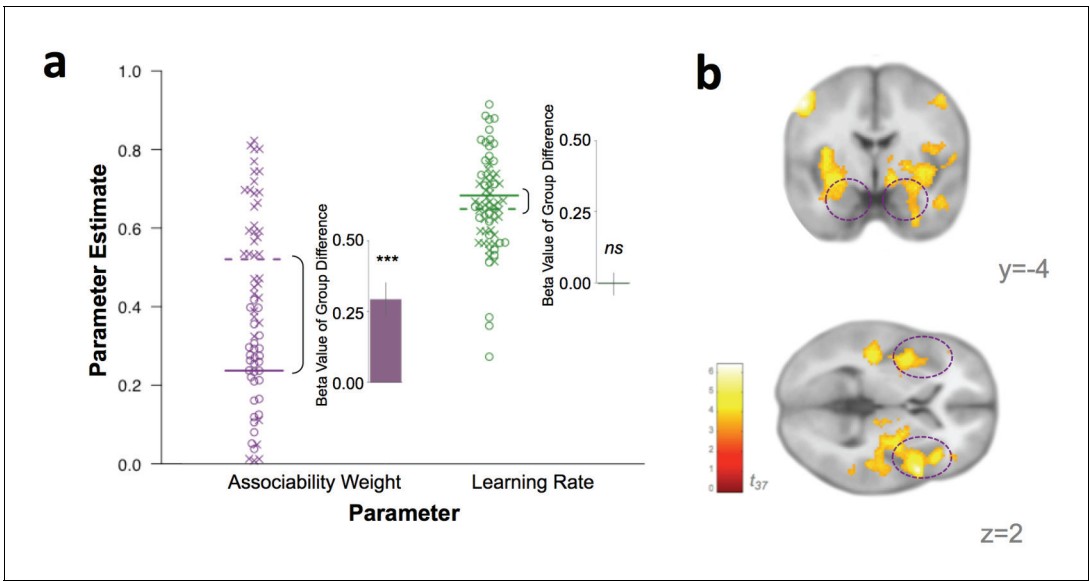

**Figure 4.** Behavioral and neural substrates of associability are increased with PTSD. (**a**) Loss associability weight increases with PTSD while unmodulated learning rate does not (individual estimates for learning parameters: circles indicate control participants [mean: solid line] and X's indicate participants with PTSD [mean: dotted line]; insets display regression beta values for PTSD diagnosis variable from linear regression predicting learning parameters; error bars represent SEM). (**b**) Neural effect of PTSD diagnosis on trial-by-trial associability value activation (cluster-level FDR p<0.05 whole brain corrected with a cluster forming threshold of p<0.001; t value of PTSD diagnosis on parametric modulator of associability value at outcome event). Dashed circles indicate amygdala and insula regions. Prediction error ROIs from an independent cohort used to test for prediction error differences in PTSD are shown in *Figure 4—figure supplement 1*.
DOI: https://doi.org/10.7554/eLife.30150.008

The following figure supplement is available for figure 4:

**Figure supplement 1.** Prediction error signaling in an independent non-trauma exposed reference cohort.
DOI: https://doi.org/10.7554/eLife.30150.009

including bilateral amygdala and insula, hypothesized areas of relevance for PTSD and associability-based learning (*Pitman et al., 2012*; *Li et al., 2011*) (*Figure 4b*; *Supplementary file 1*, Table 1C; see *Figure 3d* for associability-related signaling across participants after accounting for PTSD and covariates). To further test the localization of the increased associability-related activation in PTSD to our *a priori* hypothesized areas of amygdala and insula, we extracted beta values from these anatomical regions of interest (see Materials and methods for ROI definition). PTSD was significantly related to associability-related activation in these areas (amygdala: $t_{37} = 2.45$, p<0.05; insula: $t_{37} = 3.87$, p<0.001). Replacing the binary PTSD diagnosis with a dimensional measure of PTSD symptom severity (total CAPS score) across all veteran participants resulted in a similar pattern of effects (*Figure 5a*; *Supplementary file 1*, Table 1D). PTSD was not related to neural responses to prediction error at this whole brain level (*Figure 5—figure supplement 1a*). Follow up analyses covaried for presence of psychotropic medication, a positive screen for mild traumatic brain injury, or smoking status, and additionally tested effects within a subgroup of veterans with and without PTSD who were free from psychotropic medication and matched on estimated IQ; none of these covariates was significantly related to neural or behavioral results involving associability, and in the matched subgroup, PTSD remained related to behavioral and neural encoding of associability value (see Appendix 2, Supplemental Results for details).

The lack of whole-brain relationship between neural prediction error signals and PTSD could be due to our conservative multiple comparison correction. To assess this possibility, we further investigated the relationship between prediction error activation and PTSD within prediction error related regions of interest (ROIs) derived from a trauma-unexposed reference cohort, separate from the veteran cohort (described in Materials and methods). Neural responses in these ROIs (including ventral striatum and vmPFC) also did not show significant effects of PTSD for prediction error, even in neural regions strongly related to prediction error signaling (see Materials and methods for details; reference group prediction error activation is shown in *Figure 4—figure supplement 1*). Independent of PTSD, participants showed significant prediction error activation in striatum (left striatum: $t_{42} = 3.56$, p<0.001; right striatum: $t_{42} = 3.02$, p<.005), further supporting intact PE-related signal that is unaffected by PTSD.

To assess which PTSD symptom clusters are more associated with increased neural signaling of associability value, we examined the symptom clusters of re-experiencing, avoidance/numbing, and hyperarousal (*American Psychiatric Association, 2000*). Specifically, we tested the degree to which symptom severity in each cluster was related to neural correlates of trial-by-trial associability and prediction error, respectively. The hyperarousal and avoidance/numbing symptom clusters showed the most extensive neural responses corresponding with associability (*Figure 5a*; *Supplementary file 1*, Tables 2E and 2F), with little relationship with re-experiencing symptoms (*Figure 5a*; *Supplementary file 1*, Table 2G).

Finally, if neural encoding of associability is relevant for real-world behavioral disruptions in PTSD, the combination of PTSD and neural responsivity to associability value ought to be related to participants' likelihood of adjusting behavioral choices based on past experiences. As described above, PTSD and the hyperarousal and avoidance/numbing symptom clusters showed a strong relationship with neural activation to associability, suggesting that the interaction of PTSD and neural substrates of associability should predict switching behavior over and above neural activation to associability value alone. We therefore carried out mixed effects logistic regression models predicting switching behavior as a result of the interaction of the previous trial's outcome and associability value-related neural activation, with and without PTSD symptoms as an additional interaction term (see Materials and methods for regression model specifications). The addition of PTSD diagnosis, hyperarousal symptoms, and avoidance/numbing symptoms each significantly improved model fit for bilateral amygdala after accounting for multiple comparisons ($\alpha_{Bonferroni} = .006$; PTSD diagnosis:$\chi^2_2 = 12.5$ p<.005; hyperarousal: $\chi^2_2 = 12.6$, p<.005; avoidance/numbing: $\chi^2_2 = 12.0$, p<.005; *Figure 5b*), while re-experiencing symptoms did not ($\chi^2_2 = 6.0$, p>0.01; similar patterns were observed for bilateral insula; *Figure 5—figure supplement 1b and c*). A secondary analysis inspecting the components of this interaction revealed overall greater tendency to switch with greater associability value-related neural activation in participants with PTSD relative to controls, with additional effects of large relative to small outcomes leading to a greater tendency to switch in controls in the amygdala and insula ROIs and in PTSD in the amygdala ROI; the effect of outcomes in PTSD in the insula ROI was reversed (*Figure 5—figure supplement 2a and b*). Analogous regressions with prediction error

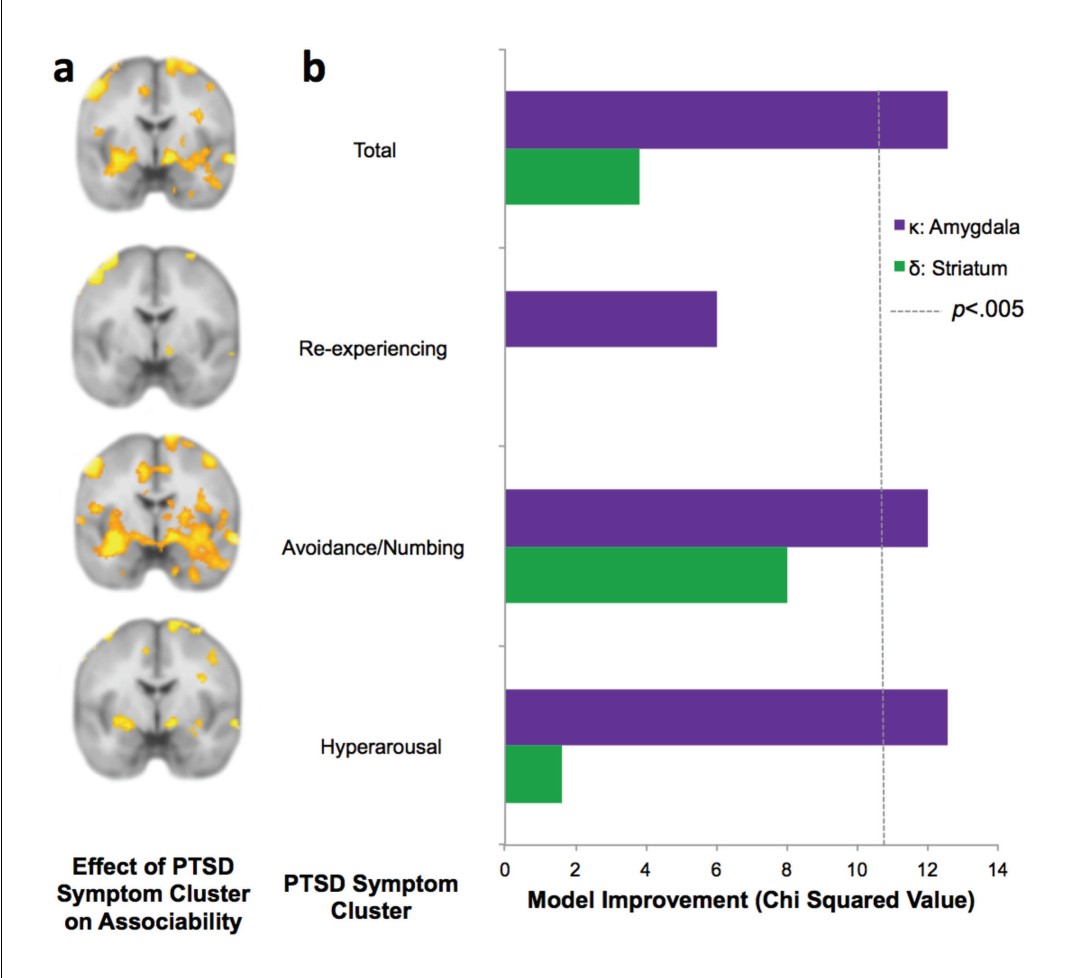

**Figure 5.** PTSD symptom clusters of hyperarousal and avoidance/numbing have greatest neural and behavioral interaction with associability value. (a) Relationship of total PTSD symptoms (CAPS score) and PTSD symptom clusters to neural encoding of loss associability value (FDR p<0.05; displayed at p<0.005 uncorrected to allow equivalent thresholding across images). (b) Switching behavior is explained by interaction of PTSD symptom clusters and neural associability value activation for total CAPS, avoidance/numbing and hyperarousal. Bars indicate improvement in model fit when adding an interaction term of symptom cluster severity to a mixed effects logistic regression model predicting switching behavior by an interaction of previous outcome with neural activity; likelihood ratio test of model fit improvement; $\chi^2$ >10.60 indicates significant improvement at p<0.005 uncorrected for multiple comparisons [$\alpha_{Bonferroni}$ =.006]). Relationship of total PTSD symptoms and symptom clusters with prediction error; associability value and prediction error analyses with vmPFC and insula ROIs; and jackknife error distributions are shown in *Figure 5—figure supplement 1*. Interaction of amygdala and insula ROIs with previous outcomes and PTSD is illustrated in *Figure 5—figure supplement 2*.
DOI: https://doi.org/10.7554/eLife.30150.010

The following figure supplements are available for figure 5:

**Figure supplement 1.** Lack of relationship between prediction error signaling and PTSD.
DOI: https://doi.org/10.7554/eLife.30150.011

**Figure supplement 2.** Plots of effects of neural responsivity to associability value.
DOI: https://doi.org/10.7554/eLife.30150.012

related activation in striatum and vmPFC were implemented to test the specificity of effects to associability; these analyses indicated that no symptom clusters improved model fit for prediction error-related neural responses predicting behavioral choices (*Figure 5—figure supplement 1b and c*).

## Discussion

Previous research has implicated altered learning and attentional processes in PTSD, but the neuromechanistic basis of these dysfunctions has been unclear. Here, we used a computational model-

based approach and identified increased associability-based modulation of loss learning, neurally and behaviorally, in combat-deployed military veterans as a function of PTSD; prediction errors and unmodulated learning rate were unchanged by PTSD. The greater neural sensitivity to associability values was particularly strong with increasing hyperarousal and avoidance/numbing symptoms and interacted with symptom severity to predict participants' behavioral choices. These results point to increased reliance on attention-based modulation of learning in PTSD that is guided by a network of brain regions including the amygdala and insula.

A number of earlier studies have connected greater attention to perceived threat, disrupted cognitive processing in the presence of negative, salient stimuli, and increased orienting to unexpected events in PTSD (*Aupperle et al., 2012*; *Bar-Haim et al., 2007*; *Blair et al., 2013*; *Morey et al., 2009*; *Naim et al., 2015*). However, the mechanistic underpinnings of this hypersensitivity to unexpected outcomes have been unclear. We posited that the hypersensitivity may derive from either (i) heightened solely error-related learning that increases learning directly from unexpected outcomes (i.e., prediction errors), or (ii) disproportionate attentional allocation to these prediction errors which magnifies the effects of these unexpected outcomes in future decision-making. Our approach differentiates between these possibilities, and the data support the latter hypothesis: neither static error-based learning rate nor neural prediction error responses were associated with PTSD, while increased associability updating based on loss prediction error signals increased with PTSD at both the behavioral and neural levels. These results suggest the disruptive effect of salient negative stimuli observed in PTSD is related to greater involvement of associability-related learning processes that are based on previous unexpected negative experiences with similar stimuli, rather than disrupted responses to immediate cues.

The current findings further link extant neural data showing amygdala and insula disruptions in PTSD (*Hayes et al., 2012*; *Brown et al., 2014*; *Rauch et al., 2006*) with the learning processes that these regions support. In particular, animal research has strongly implicated amygdala-mediated, attention-modulated processes in learning (*Esber et al., 2012*; *Holland and Gallagher, 1999*), and support for the amygdala's role in these processes is emerging in human neuroimaging studies of associative learning (*Li et al., 2011*; *Boll et al., 2013*). In addition, intriguing evidence suggests that transient lesions of amygdala functioning through direct stimulation reduces PTSD symptoms (*Langevin et al., 2016*). The insula, which here similarly demonstrates increased associability-related activation with PTSD, also plays a key role in associability-based learning (*Li et al., 2011*) and related functions including altering outcome value and computing costs (*Skvortsova et al., 2014*; *Palminteri et al., 2012*). Previously found alterations in neural connectivity and responses to affective stimuli in amygdala/insula may further derive from increases in attention- or salience-modulated learning associated with these regions that emerges as hypersensitivity to potential threat (*Etkin and Wager, 2007*; *Brown et al., 2014*; *Rauch et al., 2006*; *Seeley et al., 2007*). We note that previous work on associability-based learning in non-psychiatric participants has shown a prominent role of associability in learning from other forms of aversive stimuli, such as threat of shock (*Li et al., 2011*; *Boll et al., 2013*), which also show disruptions in PTSD (*Lissek and van Meurs, 2015*). While the increased updating of associability values observed here in PTSD during monetary loss learning should apply generally to learning from other classes of negative stimuli, it will be important in future investigations to explicitly test the role of associability-based learning for PTSD in the context of threatening or trauma-related cues which have been shown to impair fear extinction in the disorder (*Milad et al., 2009*; *Norrholm et al., 2011*) as well as to connect our findings to models of other behavioral disruptions in mood/anxiety (e.g., [*Mkrtchian et al., 2017*]) and to explore under what conditions the greater associability weight in PTSD is adaptive versus maladaptive (as in [*Vythilingam et al., 2007*; *Zinchenko et al., 2017*]). Rich literatures support associability as a measure of cue-specific salience or attention allocation (*Li et al., 2011*; *Pearce and Hall, 1980*; *Le Pelley, 2004*; *Roesch et al., 2012*), but other stimulus properties and forms of attention may also affect learning in PTSD (*Dayan et al., 2000*; *Yu and Dayan, 2005*; *de Berker et al., 2016*). In addition, as previous work has found associability-related learning in the amygdala to be dopamine-dependent (*Esber et al., 2012*), the intact error-related but disrupted associability-related signaling in PTSD found in the present study raises questions about potential intermediate neural substrates involved in this dissociation (*Nasser et al., 2017*). Future work on the relationship between error- and associability-related learning signals is needed to further clarify the neural systems-level disruptions in this relationship in PTSD.

The associations among neural encoding of associability value, specific PTSD symptoms, and behavioral choices support the translational relevance of the present findings for identifying and refining new neuromechanistically-informed targets of treatment (*Bowers and Ressler, 2015*). Here, hyperarousal and the related numbing/avoidance symptoms (*Flack et al., 2000*) showed the strongest relationships with neural substrates of associability; interactions of these neural substrates with these PTSD symptoms also predicted participants' behavioral choices during the loss learning task. The clinical manifestation of hyperarousal in PTSD includes excessive sustained vigilance (*American Psychiatric Association, 2000*) and resembles what heightened updating of associability values would predict; specifically, reminders of trauma can be conceptualized as stimuli associated with unexpected negative outcomes, with the greater associability updating seen here in PTSD causing stimuli associated with a recent history of these surprising outcomes to command greater attention and increased updating. Thus, addressing certain components of attention or cue-salience in learning-based interventions (e.g., prolonged exposure [*Craske et al., 2014*; *Schnurr et al., 2007*]) or incorporating a learning context into neural or cognitive therapies (*Badura-Brack et al., 2015*; *Langevin et al., 2016*; *Khanna et al., 2015*) may provide targeted benefit for reducing symptoms of PTSD. The increased updating of associability values in PTSD may reflect a higher prior belief that changes in prediction error will occur, a difference in processing changes in prediction errors, or a combination, which could also serve as mechanistic treatment targets. More generally, associability and other computational model-identified dysfunctions may serve as new, precise, functional targets for novel and extant treatment approaches. In particular, disruptions in associability-related attention or cue-salience neural circuits may be improved with computational model-based behavioral retraining techniques. Likewise, neural substrates of associability may be evoked during learning tasks and targeted with real-time neuromodulation.

The increased updating of associability values reported here also suggests mechanistic insight into the impaired fear extinction and retention of extinction memories that have previously been reported in PTSD (*Milad et al., 2009*; *Norrholm et al., 2011*). Specifically, greater updating of associability-modulated learning in PTSD may help explain several phenomena seen during extinction in PTSD. Greater associability signaling during cues associated with uncertain outcomes, such as during the early phases of fear extinction, would create a lack of habituation (*Norrholm et al., 2011*; *Collins and Schiller, 2013*). Meanwhile, as stimuli become more predictable, greater associability values lead to reduced learning, which may weaken retention of repeatedly presented information such as during later stages of extinction (*Craske et al., 2014*; *Milad et al., 2009*). Increased updating of associability values in PTSD may also increase the likelihood of inferring new states after surprising outcomes, rather than supporting inhibitory learning about current states, thus reducing the ability to connect conditioned and extinguished contexts and leading to impairments in extinction retention and generalization (*Dunsmoor et al., 2015*; *Morey et al., 2015*). We note that while the present work is consistent with earlier studies in non-psychiatric controls identifying associability-related neural activity at the outcome event (*Li et al., 2011*; *Esber et al., 2012*; *Roesch et al., 2010a*, *2010b*), neural substrates of associability have also been observed with predictive cues (*Roesch et al., 2012*); this distinction deserves further study in future work investigating and addressing the nature of learning disruptions in PTSD.

Despite the high prevalence and great societal, personal, and fiscal cost of PTSD, and particularly combat-related PTSD (*Richardson et al., 2010*), the mechanistic underpinnings of the disorder remain unclear. Here, a neurocomputational psychiatry approach distinguishes between competing hypotheses about dysfunctional processes in PTSD (*Montague et al., 2012*; *Stephan and Mathys, 2014*) and shows that PTSD is related to increased neural and behavioral substrates of attention-based modulation of loss learning, while solely prediction error-based learning is unchanged. More generally, these findings suggest that by integrating human functional neuroscience, computational model-based analyses, and gold-standard clinical assessments, new neuromechanistic targets of intervention for PTSD and other mental disorders may be identified and tested.

# Materials and methods

## Participants

Participants were US military veterans (N = 74) with combat deployments to Iraq or Afghanistan since 2001. All veterans had served at least one tour in Iraq or Afghanistan since 2001 and were part of a larger study by our group examining biomarkers of mood and anxiety disorders. All participants provided written informed consent, and all procedures were approved by the Institutional Review Boards of Baylor College of Medicine, the Salem Veterans Affairs Medical Center, and Virginia Tech. Veterans served by the Houston, TX and Salem, VA Veterans Affairs Medical Centers were recruited through VA medical record searches, provider referrals, and community advertisements. Enrollment inclusion criteria for the larger biomarker study included: meeting criteria for current PTSD or non-PTSD, age 18 to 64, English speaking, normal or corrected to normal vision, and verbal IQ greater than 80. Enrollment exclusion criteria included: contraindications to MRI scanning (e.g., implanted ferrous metal, claustrophobia), loss of consciousness greater than 30 min, behaviors meeting criteria for substance abuse or dependence (excluding nicotine dependence) in the past 30 days, and current/past psychotic or bipolar disorders.

For the veteran cohort, psychiatric diagnoses were assessed using the Structured Clinical Interview for DSM-IV (SCID [*First et al., 1996*]) and the Clinician Administered PTSD Scale (CAPS [*Blake et al., 1995*]), administered by trained study staff. Using the 'Rule of 3' (frequency +intensity > 3 [*Blanchard et al., 1995*]) diagnostic criterion on the CAPS. Total PTSD severity was constructed from the summed severity score for all symptoms. Severity scores for the three DSM-IV symptom clusters, measuring re-experiencing (CAPS questions B1-B5), avoidance and numbing (questions C6-C12), and hyperarousal (questions D13-D17) symptoms were created to assess the severity of each symptom cluster. Participants also completed the Beck Depression Inventory-II (BDI [*Steer et al., 1999*]) measuring depression symptoms; the Combat Exposure Scale (CES [*Lund et al., 1984*]), measuring combat trauma exposure, and the Wechsler Test of Adult Reading (WTAR [*Wechsler, 2001*]), measuring approximate verbal IQ. In addition, participants completed a demographics questionnaire assessing age, gender, ethnicity, years of education, and medications taken. Two participants did not complete the CES; their scores were imputed based on other variables (CAPS, BDI, age, and gender) using the 'mi' package in R (*Su et al., 2011*).

To be included in the present analyses, veterans were further required to demonstrate behavioral engagement on the relevant portions of the probabilistic learning task (i.e., sufficient loss learning, criteria detailed in the Probabilistic learning task section) and have successfully completed the functional MRI portion of the study. In addition, all included veteran participants were required to have experienced trauma satisfying criterion A1 of PTSD according to the DSM-IV (*American Psychiatric Association, 2000*), and veteran participants without PTSD were required to be free from current DSM-IV diagnoses and not taking psychotropic medication. N = 68 veterans satisfied these additional criteria and were considered for all subsequent behavioral analyses (see fMRI data collection and preprocessing for imaging analysis inclusion criteria). Among these 68 veterans, thirty-nine (N = 39) met DSM-IV (*American Psychiatric Association, 2000*) criteria for PTSD (see *Supplementary file 1*, Table 2A for clinical and demographic information); the other N = 29 veterans had previous deployment-related trauma exposure as assessed by the CAPS interview but did not meet criteria for PTSD. Participants excluded did not differ from those retained for all analyses on demographic measures (age, gender, household income level, proportion nonwhite, proportion with greater than high school education), PTSD severity, or proportion with a PTSD diagnosis (all ps > 0.05).

Two additional cohorts of participants were recruited from the Houston, TX and southwest Virginia areas. The first, a non-psychiatric civilian reference cohort (N = 23) was not included in the full set of analyses but was used to construct independent regions of interest to test in the veteran cohort (see fMRI procedures for more information). The second cohort of participants (N = 20) consisted of participants with a current diagnosis of Major Depressive Disorder and no current or past PTSD, matched to the participants with PTSD on gender, smoking status, and estimated IQ. Behavior from this clinical comparison cohort was fit to the associability RL model and parameter estimates were compared to the participants with PTSD (see Relationship between model parameters and PTSD for more information). The focus of the present work was on learning in PTSD; as such, given

the high co-occurrence of depression in PTSD, the MDD-only comparison group was used here to test the specificity of the main behavioral effects to PTSD.

Participants in the non-psychiatric reference cohort, used to construct independent neural regions of interest, had psychiatric diagnostic status assessed by the SCID (N = 17) or the Mini International Neuropsychiatric Interview (MINI [Lecrubier et al., 1997]; N = 6). Participants also completed the WTAR to estimate IQ and a demographics questionnaire. Participants included for ROI construction reported no trauma exposure that qualified for Criterion A1 for PTSD and were matched to the veteran cohort on age, gender, and estimated IQ. Participants in the MDD psychiatric reference cohort, whose behavioral learning parameters were compared to the participants with PTSD, had psychiatric diagnostic status assessed by the SCID and met criteria for current Major Depressive Disorder and did not meet current or lifetime criteria for PTSD. These participants also completed the BDI to assess current depression symptoms, the WTAR to estimate IQ, and a demographics questionnaire to assess age, gender, and smoking status.

## fMRI data collection and preprocessing

Participants were scanned on a 3T Siemens Tim Trio MR scanner. Echoplanar images were collected in 34 4 mm slices at a 30° hyperangulation from the anterior-posterior commissure (AC-PC) line (TR = 2000 ms, TE = 30 ms, flip angle = 90°, matrix = 64×64, voxel size 3.4 × 3.4×4.0 mm$^3$). A high-resolution (1 mm$^3$) anatomical Magnetization Prepared Rapid Gradient Echo (MPRAGE) T1 image (TR = 1200 ms, TE = 2.66 ms, flip angle = 12°) was collected to aid in registration.

All imaging analyses were conducted using SPM8 for fMRI (Wellcome Trust Centre for Neuroimaging, http://www.fil.ion.ucl.ac.uk/spm/software/spm8). Preprocessing consisted of: slice timing correction, realignment to the first functional scan, coregistration to the participant's structural image, normalization to the MNI template, and smoothing to ensure Gaussianity (6 mm FWHM). Participants with motion greater than 3 mm or 0.5 radians in any direction were excluded (n = 4). Functional images were visually inspected for signal drop out in ventral areas, including ventromedial prefrontal cortex and amygdala, and were excluded if significant signal loss was present (n = 3). An additional two participants were excluded due to other data quality issues.

## Probabilistic learning task

Participants completed a probabilistic loss and gain learning task (two-arm bandit, adapted from [Pessiglione et al., 2006]) while undergoing fMRI scanning (Figure 1). The task was presented in pseudo-randomized alternating blocks consisting of all loss learning or gain learning trials, respectively. On each trial, participants were presented with two abstract stimuli. One stimulus had a higher (75%) probability of leading to a better monetary outcome and a lower probability (25%) of leading to a worse monetary outcome, while the probabilities for the other stimulus were reversed (i.e., smaller probability of better outcome and larger probability of worse outcome). Participants selected one stimulus using a MRI-compatible button box (Current Designs, Inc.). The participant's choice was framed for a jittered viewing time of 2–4 s, after which the outcome (monetary amount gained or lost) was shown for 2 s. Participants were not shown the outcome associated with the unchosen stimulus. A fixation cross was shown between each trial for a jittered viewing time of 1–3 s. For gain trials, the better outcome ranged from +70 to +80 cents, while the worse outcome ranged from +20 to +30 cents. For loss trials, the outcomes similarly ranged from −20 to −30 and −70 to −80 cents. At the beginning of each block, high and low outcome values were randomly chosen with replacement from uniformly distributed outcome pairs {20,70}, {25,75}, or {30,80} and kept consistent within blocks (the outcome displays indicated 'You Lose [amount]' for loss blocks, and 'You Gain [amount]' for gain blocks). Blocks containing gain and loss trials alternated. Each block consisted of novel stimuli, which required participants to re-learn the contingencies between stimuli and outcomes within each block.

As our goal was to examine mechanistic processes associated with learning, including in individuals who may differ in learning processes, the task used an adaptive algorithm to ensure that participants successfully learned the contingencies and continued to learn throughout the task. In order to obtain a sufficient number of learning trials, block length was determined based on the proportion of correct choices (i.e., choosing the better option): blocks ended when performance reached at least 70% correct based on a running average of the last 10 choices, with the additional specification

that the first block be at least 15 trials long. The task ended when the participant had at least 25 correct and 25 incorrect choices in each of the loss and gain conditions. The total number of trials per participant ranged from 50 to 70 for loss trials and 50–68 for gain trials. Participants completed an average of 4.14 gain blocks and 4.58 loss blocks; the number of blocks and trials completed did not differ between PTSD and veteran control participants for gain or loss (all ps > 0.2). The comparable number of trials and blocks completed in each group indicates that the amount of task titration was similar on average between the PTSD and veteran control participants.

Before entering the scanner, participants were presented with task instructions, completed a practice round, and were given the opportunity to ask questions. They were not provided information about the full statistical structure of the task but were provided an initial $10 endowment, and informed that payment would be based on actual performance. To ensure participants were attending to the task and had behavior suitable for model fitting (*Sokol-Hessner et al., 2009*), participants who switched options in either gain or loss blocks less than 10% of the time were excluded from analyses comparing gain and loss behavior (N = 19; 8 control veterans and 11 veterans with PTSD). For analyses of loss behavior only, participants excluded for low switching during gain blocks only were included in analyses (see Comparison of model fits for RL models for more information).

## Reinforcement learning models

To identify the reinforcement learning (RL) model that best explained participants' data, models with and without associability were tested; models were evaluated separately for gain and loss trials.

Following standard temporal difference RL (*Sutton and Barto, 1998*) the expected value (Q) of the stimulus on the next trial (t + 1) was updated with the product of the learning rate (α; range 0 to 1) and prediction error (δ):

$$\mathbf{Q}_A(t+1) = \mathbf{Q}_A(t) + \alpha * \delta(t) \tag{1}$$

Trial-by-trial prediction error $\delta(t)$ was computed as the difference between the actual outcome (**R**) and the expected value (**Q**):

$$\delta(t) = \mathbf{R}'(t) - \mathbf{Q}(t) \tag{2}$$

The outcome **R** was modulated by reward sensitivity (ρ) (*Huys et al., 2013*), comprising a multiplier on the value of the loss or gain value further from 0 (i.e., outcomes ranging ±0.70 to ±0.80):

$$\mathbf{R}'_A(t) = \rho \mathbf{R}_A(t) \tag{3}$$

where A represents the chosen option. A decay parameter for the unchosen option B (γ; range 0+) (*Boorman et al., 2009*; *Collins and Frank, 2016*; *Niv et al., 2015*; *Cavanagh, 2015*) was also included:

$$\mathbf{Q}_B(t+1) = \gamma \mathbf{Q}_B(t) \tag{4}$$

The probability of each choice was modeled with a softmax function incorporating inverse temperature (β; range 0+):

$$\mathbf{P}_A(t+1) = \frac{e^{\beta \mathbf{Q}_A(t+1)}}{e^{\beta \mathbf{Q}_A(t+1)} + e^{\beta \mathbf{Q}_B(t+1)}} \tag{5}$$

Since inverse temperature and reward sensitivity may not be uniquely identifiable in the RL model, inverse temperature was first estimated in a model without reward sensitivity. The RL model with reward sensitivity was then estimated with inverse temperature fixed at the group mean rather than estimated as a free parameter (similar to [*Pessiglione et al., 2006*]).

For the associability RL model, learning rate was modulated on a trial-by-trial basis by an associability value for the chosen stimulus (κ) (*Li et al., 2011*). An associability weight parameter (η; range 0 to 1) controlled the extent to which the magnitude of previous prediction errors updated the trial-by-trial associability value of a particular stimulus. Associability values were initialized at one and updated for each stimulus separately. Associability values were constrained to stay above. 05 (*Li et al., 2011*; *Le Pelley, 2004*) . Therefore, similar to the constant parameter of learning rate and the trial-by-trial estimate of prediction error in classic RL models, this model added a constant

parameter of associability weight and a trial-by-trial estimate of associability value of the chosen stimulus.

$$\kappa_A(t+1) = (1-\eta)\kappa_A(t) + \eta|\delta(t)| \tag{6}$$

$$\mathbf{Q}_A(t+1) = \mathbf{Q}_A(t) + \alpha * \kappa_A(t) * \delta(t) \tag{7}$$

Regressors for imaging analyses used the prediction error, associability value, and probability at trial t, prior to updating associability value for the next trial. Because novel stimuli were used each block, the expected value and associability value were reset at the beginning of each block. The same parameters were used for all blocks of the same condition, but separate parameters were used per condition, resulting in one learning rate for loss and one learning rate for gain, one associability weight for gain and one for loss, and so on.

## Comparison of model fits for RL models

Model comparison was computed using Bayesian Model Selection (*Rigoux et al., 2014*). Each participant's probability for each model was transformed into corrected AIC (AICc) to penalize models with greater numbers of parameters. These values were then fed into a variational approximation of BMS, resulting in a predicted probability of each model per subject and an overall protected exceedance probability of each model over all subjects. This approach allows for between-subject heterogeneity in model probabilities while enabling a group-level model comparison.

Reinforcement learning model parameters were estimated separately for loss and gain trials, based on previous work showing differential influences of reward and punishment during learning in similar tasks (*Palminteri et al., 2012*; *Pessiglione et al., 2006*). To test the appropriateness of this approach, we first conducted a logistic regression predicting the probability of switching choices based on the previous three outcomes, including the condition (loss or gain) as an interaction term on the previous outcome. This interaction term was significant (z = 2.124, p<0.05), suggesting different learning patterns during loss and gain learning. As a second test of the appropriateness of separate estimates for loss and gain, we compared a model with parameters combined across loss and gain trials to a model separating all parameters by condition. The model with all parameters separated by condition fit better (AICc improvement in 91% of participants, protected exceedance probability of 100%), confirming different learning patterns in loss and gain conditions. Additionally, we tested the fit of a basic reinforcement learning model, with parameters of learning rate and inverse temperature, versus our model with parameters of learning rate, reward sensitivity, and decay (prior to adding the additional parameter of associability weight) and found that the more complex model fit better, with a protected exceedance probability of 100%.

Reinforcement learning models with and without associability were then compared within loss and gain conditions separately. 81% of participants (85% of veteran controls and 78% of PTSD participants) showed a greater probability of the model with associability in loss, while no participants showed a greater probability of the associability model during gain learning. This difference in frequencies between conditions was significantly different ($\chi^2_1$ = 59.0, p<0.001). The overall protected exceedance probability of the associability model reflected this difference, with a protected exceedance probability of 100% in loss and 0% in gain (*Figure 3a*). As a result of the improved model fit in the loss condition only, subsequent analyses focused on loss learning trials. To maximize power, participants with suitable loss behavior but a low rate of switching during gain learning (n = 16) or who were excluded from imaging analyses (n = 9) were included in model-based behavioral analyses, resulting in a sample size of 68 (39 PTSD and 29 veteran controls); analyses comparing loss and gain learning used the more restricted sample of 43 participants (23 PTSD and 20 veteran controls) with complete data across conditions and modalities (i.e., loss, gain, behavior, neuroimaging). Analyses of these participants with full data in all four categories are reported in Appendix 2, Supplementary Results, Model-based loss analyses in restricted sample and are consistent with results from the larger group.

## Model testing and validation

To delineate the effect of associability weight on performance accuracy, we conducted a simulation of participants' performance at different values of the associability weight. Other parameters were

set at the group means and the associability weight was varied from. 05 to. 95 in steps of. 05. Data from fifty participants with two blocks of 25 trials each were simulated for each value of the associability weight. The mean and standard error of the proportion of correct options chosen were calculated for each value of associability weight. Performance did not systematically vary with the value of the associability weight and the correlation between associability weight and performance was not significant (r = 0.10, p>0.1; *Figure 3—figure supplement 1a*).

According to theories of associability-based learning (*Le Pelley, 2004*), stimuli that have been poorly predicted in the past should draw more attention during subsequent learning trials, resulting in increased associability values and longer decision times. Therefore, we tested whether reaction times scaled with the associability value of the chosen cue. Each participant's reaction times were modeled in a regression equation with predictors of the chosen cue's associability value, the chosen cue's expected value, and the trial number; trials with reaction times shorter than 333 milliseconds or longer than five seconds were excluded (3.07% of trials). *Figure 3b* shows participants' average beta values of the relationship between associability value and reaction time during loss learning. The average beta value was .232, and significantly greater than 0 (one sample t-test: $t_{68}$ = 8.28, p<0.001), reflecting longer decision times for stimuli with larger associability values. The average beta value for associability value predicting reaction time when including the previous trial's prediction error as an additional predictor was similarly significant ($t_{68}$ = 7.21, p<0.001), indicating that the effect of associability value on reaction time was not due to concomitant prediction error. This effect was confirmed in a mixed effects regression using the 'lme4' package (*Bates et al., 2015*) in R including all participants' reaction times and accounting for the nesting of responses within subjects. Model fit was compared between a model predicting reaction times from the chosen cue's associability value, the chosen cue's expected value, and trial number versus a model without associability value; the model with associability value was a significantly better fit (likelihood ratio test: $\chi^2_1$ = 8.67, p<.005). Trial number was included as a covariate in these analyses to account for general adaptation effects; however, trial number and associability were uncorrelated (average correlation across subjects: $r$ = −0.12).

To investigate the associability RL model's performance in predicting participants' behavior, each participant's trials were binned (using 5 bins total) based on the probability of choosing the correct choice as predicted by the model. This binning resulted in a proportion of correct choices made when the predicted probability of choosing the correct stimulus was 0–20%, 31–40%, and so on per participant. The mean and standard error of the proportion of correct choices per bin was calculated across subjects. The model showed good concordance with actual behavior; binned predicted performance was strongly correlated with mean proportion of correct choices per bin ($r_4$ = 0.9986, p<0.001; *Figure 3—figure supplement 1b*).

As an additional test of the ability of the RL model's associability value to predict behavior in the loss condition, mixed effects logistic regression models were used to assess the effects of prediction error alone and prediction error modulated by associability value on model free switching behavior using the 'lme4' package in R. The dependent variable was whether or not the participant switched choices from the previous trial to the current trial (0 = no switch, 1 = switch). The basic regression included terms for the previous trial's outcome (1 = small loss, 0 = large loss) and for the nesting of trials within subjects (e.g., switch ~ outcome 1 trial previous + [1|subject]). The fit (AIC) for this model was compared against two more complex models: (1) adding an interaction of the previous trial's outcome with the previous trial's prediction error (e.g., outcome 1 trial previous * PE 1 trial previous) and (2) adding an interaction of the previous trial's outcome, prediction error, and associability value (outcome 1 trial previous * PE 1 trial previous * associability value 1 trial previous). Prediction error and associability value estimates were taken from the associability RL model using individual parameter estimates. The difference in AIC between the basic regression model and models adding prediction error and prediction error * associability is plotted in *Figure 3c*. Standard error on the improvement in AIC was calculated using a jackknife approach (*Efron and Gong, 1983*).

## Relationship between model parameters and PTSD

The relationships between PTSD diagnosis and individual behavioral parameter estimates for associability weight and learning rate were computed in R using a linear regression with additional regressors of BDI, CES, age, and gender. The variance inflation factor for PTSD in this analysis was within acceptable limits (VIF = 1.56). The beta values for PTSD diagnosis (control vs. PTSD; error bars of

standard error) predicting associability weight and learning rate, respectively, are plotted in *Figure 4a*.

As an additional test of the specificity of these findings to PTSD versus general psychopathology, RL parameters from the behavior of an additional psychiatric reference cohort with diagnoses of MDD but not PTSD (see **Participants**, above) were compared to the participants with PTSD, covarying for age. A linear regression model was used to compare these groups accounting for age; a regression with additional covariates of gender, estimated IQ, smoking status, and depression severity showed similar results supporting the specificity of increased associability to PTSD (PTSD diagnosis effect: $t_{53} = 7.29$, $p<0.001$).

## fMRI Analyses

For first level analyses, analyses of prediction error used prediction error ($\delta$) as a parametric modulator at the time of outcome. Based on previous work showing electrophysiological and BOLD response to associability at time of outcome (*Li et al., 2011*; *Esber et al., 2012*; *Roesch et al., 2010a*), associability analyses used associability value ($\kappa$) as a parametric modulator on the outcome event. An additional analysis combined these parametric modulators in the same first-level analysis, with prediction error entered as the first parametric modulator and associability value as the second; this approach did not meaningfully differ from entering these parametric modulators in separate first-level analyses (see *Supplementary file 1*, Table 1H for second level results for this approach). All analyses used the probability of the chosen option (computed by the softmax function) as a parametric modulator at time of cue presentation and a second parametric modulator of outcome amount at the time of outcome. All outcome and cue events were modeled as stick functions. Prediction error, associability value, and probability were computed based on each subject's individually estimated parameters and were z-transformed prior to entering at the first level (*Lebreton and Palminteri, 2016*). Additional regressors of no interest included button press, block number, and six motion regressors. Data were high pass filtered with a cutoff of 128 s. Group level analyses were set to a voxel-level cluster forming threshold of $p<0.001$ and then cluster-level corrected at $p<0.05$ topological false discovery rate (FDR) for the whole brain (*Woo et al., 2014*).

To assess the impact of PTSD (or symptom severity) on neural substrates of prediction error and associability, PTSD diagnosis or Clinician Administered PTSD Scale (CAPS) symptom cluster score was entered at the group level with covariates of depression, combat exposure, age, and gender. Follow-up analyses included the presence of psychotropic medication and mTBI as covariates (see Appendix 2, Supplementary Results, Subgroup Analyses for details). To test the robustness of our results to the thresholding method used, the effect of PTSD diagnosis on associability-related neural signal at the second level was checked using a nonparametric thresholding approach. Using SnPM13 (version 1.05; http://warwick.ac.us/snpm), 10,000 permutations were carried out with PTSD as the covariate of interest and depression, combat exposure, age, and gender as covariates of no interest. Results were thresholded at $p<0.05$ FWE, resulting in a corrected threshold of $t > 5.36$. Results with this permutation analysis were similar to those reported in the main text and are displayed in *Supplementary file 1*, Table 1I.

To test whether the lack of evidence for a relationship between neural prediction error and PTSD severity was due to conservative whole-brain correction, we followed up with a region of interest analysis in areas involved in prediction error. To select areas most likely to be involved in prediction error, functional regions of interest (ROIs) were created based on activations from the probabilistic learning task in a non-veteran reference cohort (see Participants, above). Group-level activations for prediction error and associability were thresholded at $p<0.005$ uncorrected with a cluster extent $\geq 20$ voxels. Activation clusters were binarized as masks and used to create ROIs (21 for gain PE, 13 for loss PE). The first eigenvariate of the beta values in each ROI was extracted for the veteran group. Values were read into R and regressed against PTSD diagnosis along with covariates of BDI, CES, age, and gender. To verify intact prediction error signaling, ROIs in the striatum were used in one-way t-tests to determine if mean activation for the veteran cohort was significantly different from zero.

## Neural interaction predicting behavioral switching

To evaluate the relevance of neural substrates of associability to behavioral disruptions in PTSD, we tested whether interactions among PTSD, neural activation in areas related to associability or prediction error, and previous outcomes predicted switching behavior. The anatomically defined regions of interest were created in the following manner: amygdala from centromedial and laterobasal subregions from the Jülich atlas (*Amunts et al., 2005*); striatum from caudate, putamen, and globus pallidus from the AAL atlas (*Tzourio-Mazoyer et al., 2002*) and nucleus accumbens from the IBASPM atlas in WFU PickAtlas; anterior insula from portion of insula anterior to y = 10 from the AAL atlas; and ventromedial PFC from medial orbitofrontal cortex from AAL. Each subject's beta values for associability value-related activation (in amygdala and insula) and prediction error-related activation (in striatum and vmPFC) were extracted from the respective ROIs. The regression included terms for the interaction of the ROI beta values with PTSD diagnosis or symptom cluster severity and the previous trial's outcome (1 = small loss, 0 = large loss) as well as the outcomes of two and three trials previous and for the nesting of trials within subjects (e.g., switch ~ previous trial's outcome * associability-related amygdala beta value * PTSD diagnosis + outcome 2 trials previous + outcome three trials previous + [1|subject], plus all lower order terms). The effects of the three previous outcomes were included in the regression as they were all found to affect switching behavior. Models were estimated using the 'lme4' package in R. Likelihood ratio tests compared this model to a model with the same ROI, but without the PTSD terms (main effect and interaction). Changes in model fit, rather than the significance of individual regressors, was chosen due the unreliability of p-values in mixed effects models such as the one used here (*Snijders, 2011*). Therefore, improvement in model fit with the addition of a term is analogous to this term being significant in the regression model. Alpha level was set at p<0.006 (Bonferroni corrected for 8 tests). *Figure 5b* shows the chi squared value from these likelihood ratio tests for amygdala and striatum, with chi squared values corresponding to p<0.005 noted for reference. Analyses with all ROIs are shown in *Figure 5—figure supplement 1b*. Although the dependent and independent variables in this analysis (switching behavior and associability-related neural activation) are related to behavioral associability parameters, this relationship reflects the underlying theorized relationship among behavior, the hybrid RL model, and neural activation rather than representing a circular analysis.

To assess subject-level variability in the likelihood ratio tests, jackknife analyses (*Efron and Gong, 1983*) were run on each hierarchical logistic regression. The distribution of the chi-squared values are shown in *Figure 5—figure supplement 1c*. Supporting the main analyses (*Figure 5*), the interaction of associability-related neural activation in amygdala with PTSD diagnosis, hyperarousal symptoms, and to a lesser extent avoidance/numbing symptoms, all showed the majority of jackknife chi squared estimates above the corrected significance levels. Meanwhile, the interaction of associability-related neural activation in amygdala with re-experiencing, as well as prediction error-related neural activation in striatum with PTSD diagnosis and all symptom clusters, all showed the majority of estimates below significance.

## Acknowledgements

This work was supported in part by the National Institutes of Health (MH087692, MH106756 to PC), the Department of Veteran Affairs (D7030R to BKC), and the National Natural Science Foundation of China (31671171, 31630034 to LZ). We acknowledge Read Montague and the assistance of Rizwan Ali, George Christopoulos, Dongil Chung, Alec Solway, Jessica Eiseman, Katherine Gardner, David Graham, Jacob Lee, Katherine McCurry, Robert McNamara, Cari Rosoff, Dharol Tankersley, and Wright Williams.

## Additional information

### Funding

| Funder | Grant reference number | Author |
|---|---|---|
| National Natural Science Foundation of China | 31671171 | Lusha Zhu |

| National Natural Science Foundation of China | 31630034 | Lusha Zhu |
| Department of Veteran Affairs | D7030R | Brooks King-Casas |
| National Institutes of Health | MH087692 | Pearl H Chiu |
| National Institutes of Health | MH106756 | Pearl H Chiu |

The funders had no role in study design, data collection and interpretation, or the decision to submit the work for publication.

### Author contributions

Vanessa M Brown, Software, Formal analysis, Validation, Visualization, Methodology, Writing—original draft, Writing—review and editing; Lusha Zhu, John M Wang, Software, Visualization, Methodology, Writing—review and editing; B Christopher Frueh, Conceptualization, Supervision, Project administration, Writing—review and editing; Brooks King-Casas, Conceptualization, Resources, Formal analysis, Supervision, Funding acquisition, Visualization, Methodology, Project administration, Writing—review and editing; Pearl H Chiu, Conceptualization, Resources, Formal analysis, Supervision, Funding acquisition, Visualization, Methodology, Writing—original draft, Project administration, Writing—review and editing

### Author ORCIDs

Vanessa M Brown (iD) http://orcid.org/0000-0003-0284-0890
Lusha Zhu (iD) http://orcid.org/0000-0001-8717-6356
Pearl H Chiu (iD) http://orcid.org/0000-0002-8767-7406

### Ethics

Human subjects: All participants provided written informed consent. Study procedures were reviewed and approved by the Institutional Review Boards of Baylor College of Medicine, the Salem Veterans Affairs Medical Center, and Virginia Tech (protocols H-22764, H-21660; BKC-0001; 15-965, 10-1044).

### Decision letter and Author response

Decision letter https://doi.org/10.7554/eLife.30150.020
Author response https://doi.org/10.7554/eLife.30150.021

## Additional files

### Supplementary files

• Supplementary file 1. Supplementary Tables 1A through 1I. (A) Table 1A. Demographic and clinical characteristics of veteran sample (N = 68). Table 1A legend: [a]mean ± SE; t-test; [b]number (%); $\chi^2$ test. CAPS: Clinician Administered PTSD Scale; BDI: Beck Depression Inventory; CES: Combat Exposure Scale; WTAR: Wechsler Test of Adult Reading. (B) Table 1B. fMRI activation clusters for loss associability independent of covariates of PTSD, BDI, CES, age, and gender (intercept term; associated with *Figure 3d*). (C) Table 1C. fMRI activation clusters for loss associability correlation with PTSD diagnosis; covariates of BDI, CES, age, and gender (associated with *Figure 4b*). Table 1C legend: Italicized peaks are local maxima in separate anatomical regions > 4 mm apart. (D) Table 1D. fMRI activation clusters for loss associability correlation with PTSD symptom severity (total CAPS score); covariates of BDI, CES, age, and gender (associated with *Figure 5a*). Table 1D legend: Italicized peaks are local maxima in separate anatomical regions > 4 mm apart. (E) Table 1E. fMRI activation clusters for loss associability correlation with hyperarousal; covariates of BDI, CES, age, and gender (associated with *Figure 5a*). (F) Table 1F. fMRI activation clusters for loss associability correlation with avoidance/numbing; covariates of BDI, CES, age, and gender (associated with *Figure 5a*). Table 1F legend: Italicized peaks are local maxima in separate anatomical regions > 4 mm apart. (G) Table 1G. fMRI activation clusters for loss associability correlation with re-experiencing; covariates of BDI, CES, age, and gender (associated with *Figure 5a*). (H) Table 1H. fMRI

activation clusters for loss associability correlation with PTSD diagnosis with associability value entered as a second parametric modulator after prediction error; covariates of BDI, CES, age, and gender (associated with *Figure 4b*). (I) Table 1I. fMRI activation clusters for loss associability correlation with PTSD diagnosis; covariates of BDI, CES, age, and gender, thresholded FWE p<0.05 using permutation testing (associated with *Figure 4b*)

DOI: https://doi.org/10.7554/eLife.30150.013

• Transparent reporting form

DOI: https://doi.org/10.7554/eLife.30150.014

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

# Appendix 1

DOI: https://doi.org/10.7554/eLife.30150.015

## Supplementary methods

### Model-agnostic behavioral analyses

Model-agnostic analyses assessed proportion of correct choices (i.e., better option chosen) between veteran participants with and without PTSD. To investigate associability-based learning in PTSD in a model-agnostic way, we conducted logistic regressions predicting participants' probability of switching based on the previous outcome as well as the unexpectedness of prior outcomes. To quantify the amount of surprise or unexpectedness in a manner similar to our model-based analyses, we defined the expected outcome in terms of the history of outcomes to that point in the block, such that the expected outcome of a stimulus equaled the proportion of time the chosen stimulus had led to the better outcome in the block or the other stimulus had led to the worse outcome. For example, if stimulus A had been picked three times, with two better outcomes, and stimulus B had been picked two times, with one worse outcome, the outcome history for stimulus A would be (2 + 1)/5 = 0.6. In addition, to account for the fact that each stimulus carried an associability value that was updated only when that stimulus was chosen, the amount of surprise was calculated anew each time a stimulus was chosen, and carried forward from the previous trial when it was not. The interaction of expectedness of the outcome two trials back with the value of the most recent outcome was prominent in loss trials ($z = -3.08$, p=0.002; gain trials: $z = -1.74$, p=0.082), supporting the role of associability-based learning in the loss domain. When examining each group separately during loss learning, the interaction of previously unexpected outcomes and the most recent outcome was more evident in the PTSD participants ($z = -3.02$, p=0.003; control participants: $z = -1.37$, p=0.17), supporting greater associability-based learning in participants with PTSD during loss learning.

### Parameter estimation and recovery

Reinforcement learning models with and without associability were fit using hierarchical Bayesian estimation (*Daw, 2011*; *Wiecki et al., 2015*). Estimation was done using Hamiltonian Markov Chain Monte Carlo using Stan (*Carpenter, 2017*) in R. Group level parameters were specified as normally distributed, with an inverse logit transformation on parameters constrained to be between 0 and 1 (learning rate and associability weight) and a lower bound of 0 on parameters constrained to be positive (reward sensitivity and inverse temperature). Parameters were given a non-centered parameterization to aid in estimation by specifying mean, scale, and error distributions for each parameter (*Betancourt and Girolami, 2015*). Mean distributions, estimated at the group level, were specified as normally distributed with priors of mean = 0 and standard deviation = 10, for parameters that are not logit transformed, or with standard deviation = 2.5 for parameters that were logit transformed. Scale distributions, estimated at the group level, were given a half-Cauchy prior (*Gelman, 2014*) (bounded to be greater than 0) with values of 0 and 2.5. Error distributions, which were estimated for each subject, were given a normal prior with mean = 0 and standard deviation = 1. For learning rate and associability weight, these parameter values were then run through an inverse logit transformation bound between 0 and 1. Therefore, each subject's parameter (for example, associability weight) consisted of a group estimated mean value plus the combined value of the group estimated scale value multiplied by the individually estimated error value. Models were estimated separately for each condition and within each veteran group. Four MCMC chains were run for each RL model, with 8000 samples per chain (4000 after discarding warm up samples). Chains from the RL models used in analyses were inspected for convergence and showed good mixing, with all values of the potential scale reduction factor (*Gelman and Rubin, 1992*) less than 1.1.

In the loss condition, the fit parameters for the veteran controls (inverse temperature fixed at 4.35) were: associability weight = 0.246 (SE. 033), learning rate = 0.708 (SE. 020), reward sensitivity = 1.06 (SE. 065), and decay of unchosen option = 0.857 (SE. 022) and for the PTSD subjects (inverse temperature fixed at 5.05): associability weight = 0.540 (SE. 039), learning rate = 0.672 (SE. 011), reward sensitivity = 1.06 (SE. 069), and decay of unchosen option = 0.883 (SE. 010). As reported throughout, the RL model with associability weight did not fit well in the gain condition (see Comparison of model fits for RL models section above); for completeness, we also report the parameters of the RL model without associability weight in gain: for the veteran controls (inverse temperature fixed at 6.35): learning rate = 0.10 (SE = 0.02), reward sensitivity = 3.01 (SE = 0.09), and decay of unchosen option = 0.82 (SE = 0.01); and for the PTSD subjects (inverse temperature fixed at 7.20): learning rate = 0.10 (SE = 0.02), reward sensitivity = 3.18 (SE = 0.09), and decay of unchosen option = 0.89 (SE = 0.01).

To test whether model parameters were independently recoverable and did not covary in a way that would prevent accurate estimation of each parameter separately, data were simulated based on different values of parameters along the range seen in participants' data (learning rates of. 67,. 75, and. 83; associability weights of. 25,. 375, and. 5; reward sensitivity of 1.0, 1.25, and 1.5; and decay of unchosen option of. 9,. 95, and 1). These simulated data were then fit to the model and the resulting recovered parameters were compared to the simulated parameters. All parameters were recoverable, with simulated differences in parameter values similar to recovered differences (*Figure 3—figure supplement 1c*).

## Appendix 2

DOI: https://doi.org/10.7554/eLife.30150.016

## Supplementary results

### Gain learning

As reported in the main text, model fit comparison showed a model with an associability parameter to be a worse fit for the gain condition. The presence of associability-based learning during exclusively negative contexts is supported by previous studies on the topic (*Li et al., 2011*; *Roesch et al., 2012*; *Boll et al., 2013*).

Similar to loss prediction error, striatal ROIs showed significant activation with prediction error independent of PTSD symptoms ($t_{42} = 2.87$; $p<0.01$), confirming intact PE-related signal in the veteran cohort during gain learning.

### Subgroup analyses (depression, combat exposure, medication, traumatic brain injury, smoking, IQ)

#### Depression and combat exposure

The covariates of depressive symptoms (as measured by BDI) and level of combat exposure (as measured by CES) were not related to variables of interest on a behavioral or neural level. Behavioral associability weight was not related to depressive symptoms ($t_{62} = 0.31$, $p>0.1$) or to level of combat exposure ($t_{62} = 0.88$, $p>0.1$). In addition, when depression diagnosis (from the SCID) was substituted for level of depressive symptoms in analyses, it also did not show a relationship with behavioral associability weight ($t_{62} = -1.51$, $p>0.2$). Level of depressive symptoms, depression diagnosis, and level of combat exposure all did not show a significant relationship with fMRI regressions against associability value.

#### Psychotropic medications and traumatic brain injury

Covariates accounting for the use of psychotropic medications or the presence of mTBI were not related to variables of interest on a behavioral or neural level. Behavioral associability weight was not related to the use of psychotropic medication ($t_{61} = 0.43$, $p>0.2$) or to the presence of mTBI ($t_{61} = -1.14$, $p>0.2$). For fMRI regressions of associability value against medication or mTBI status, no significant relationships were observed ($p<0.001$ cluster-forming threshold).

#### Smoking

Three out of the 29 veteran controls without PTSD and eight of the 39 veterans with PTSD reported being a current cigarette smoker. The chi-squared test for differences in frequencies of cigarette smokers in these two groups was not significant ($\chi^2_1 = 0.692$, $p>0.2$). We additionally investigated smoking status as a covariate in a series of follow-up analyses to determine if nicotine consumption was related to our effects of interest. Smoking status was not related to loss associability or learning rate (associability: $t_{61} = 1.20$, $p>0.2$; learning rate: $t_{61} = -0.18$, $p>0.2$) and including this variable did not meaningfully alter the relationship between PTSD diagnosis and these dependent variables (relationship with PTSD diagnosis: learning rate $t_{61} = -0.004$, $p>0.2$; associability $t_{61} = 4.50$, $p<0.001$). Smoking status also did not show any significant relationship with neural associability, and including this variable in the neural analyses did not meaningfully change the relationship between PTSD status and associability-related activation.

#### IQ

Since lower intelligence is a risk factor for PTSD, and has been widely documented to differ between those who develop PTSD after a trauma and those who do not (*Breslau et al., 2006*; *Koenen et al., 2007*), IQ was not included as a covariate in the primary analyses. When

including estimated verbal IQ (Wechsler Test of Adult Reading score) as a covariate, the relationship between estimated verbal IQ and behavioral associability weight is nonsignificant ($t_{61} = -0.94$, p>0.2) as well as with behavioral learning rate ($t_{61} = 1.93$, p>0.05), while neural effects of associability do not show a relationship with IQ. Meanwhile, the effect of PTSD behaviorally and neurally remains significant when including estimated IQ as a regressor.

## Subgroup matched on IQ and medication

As an additional test of the robustness of effects in PTSD to the potential confounds of estimated IQ and medication status, a subgroup of participants who were both (1) free from psychotropic medication and (2) matched between PTSD and veteran control groups on estimated IQ was analyzed. The effect of PTSD on behavioral associability weight ($t_{29} = 2.99$, p<0.01) and neural associability signal in ROIs of amygdala ($t_{20} = 2.74$, p<0.05) and insula ($t_{20} = 4.42$, p<0.001) remained significant in this subgroup.

## Model-based loss analyses in restricted sample

As reported above, our initial analyses included any participants with suitable loss behavior (N = 68). We repeated these behavioral analyses in the subset of veteran participants with suitable behavior and imaging data for both loss and gain trials (N = 43; 23 patients with PTSD and 20 veteran controls). Mirroring the findings in the full set of participants, participants with PTSD in this smaller sample had a higher behavioral associability weight ($t_{37} = 2.86$, p<0.01) and no difference in learning rate ($t_{37} = -0.45$, p>0.2), an effect that was robust to additional covariates of medication, smoking status, and mild TBI.

