## [Decision Letter]

Thank you for submitting your article "Associability-modulated loss learning is increased in posttraumatic stress disorder" for consideration by *eLife*. Your article has been reviewed by two peer reviewers, and the evaluation has been overseen by a Reviewing Editor and Richard Ivry as the Senior Editor. The reviewers have opted to remain anonymous.

The reviewers have discussed the reviews with one another and the Reviewing Editor has drafted this decision to help you prepare a revised submission.

Summary:

This study addresses the question of how the hypervigilance in PTSD patients can be accounted for from a computational psychiatric perspective. The authors provided both behavioral modeling and neuroimaging data to support the hypothesis that allocation of attention (associability weight) is higher in the PTSD compared to the control subjects; neural markers associated with associability but not prediction error, such as amygdala and insula showed elevated activity in PTSD during loss learning. This is a well-written and timely research article with interesting results. The quantitative approach to a specific clinical population should be of interest to a wide audience of basic research and translational studies.

Essential revisions:

1) Both reviewers noted that the modeling part was potentially more complex than needed. For example, the paper specified reward sensitivity and learning rate. However, these two parameters might covariate in parameter estimation. They also separated the chosen and unchosen option learning rates yet did not run a nested model comparison to test whether these additional parameters are necessary. From an empirical perspective, the authors might want to restrict their model to a simpler one or provide enough evidence to support a more complicated one. Also, Equation 3 does not seem to make sense as currently written.

2) Please add some additional detail regarding the task to the main text. It is difficult to understand the model without understanding the task. Within the task description in the supplemental materials, several aspects of the task remain unclear:

- What was the probability of better/worse outcomes for the less rewarded stimulus?

- What was the probability distribution of payoffs within the specified ranges, i.e. within 20-30 or 70-80 (uniform, Gaussian, etc.)?

- Were outcomes shown for both chosen and unchosen stimuli? If not, how did subjects learn from the unchosen stimulus?

3) The authors might want to clarify the logic behind titrating the length of the task to achieve certain level of better choice selection rates. Different performance in the task itself is a behavioral marker between the control and PTSD groups. In fact, Figure 2 suggests that there is a difference in performance between the groups. Also, given the design of the task, gain and loss blocks alternated. Within the gain block, associability in general shows a declining trend. Thus, it is important to disentangle a non-specific adaptation signal from the associability signal that the paper focuses on.

4) There are significant problems with the conceptual interpretation of the results:

-Based on the equations, associability value is expected (unsigned) prediction error:

Equation 6 can be rewritten as 𝛋A(t + 1) = 𝛋A(t) +η(|𝛅 t | – 𝛋A(t))

Associability weight is the learning rate in learning this associability value (i.e. in learning expected unsigned prediction error). Presumably, this learning rate should be high if the prediction errors themselves are expected to be more likely to change over time.

If this interpretation is correct, many descriptions of associability weight in the text are misleading. For example, "behavioral parameter estimates of associability…increased with PTSD" (Abstract), "[Associability weight] indicates the extent to which the magnitude of previous prediction errors is used to update trial-by-trial associability values", "associability weights, reflecting the degree to which stimulus values are modulated or not by associability during loss learning for each participant". Associability weight is actually a learning rate and so reflects the extent to which recent prediction errors (as opposed to older prediction errors) are weighted in updating associability values. This is different from stating associability weight is a measure of associability or that it reflects the extent to which associability is used to make decisions. This distinction is crucially important to understanding the results conceptually.

5) It seems that associability does not play a stronger role in subjects with PTSD. It also does not appear to be the case that PTSD subjects are systematically overestimating associability values. Rather, subjects with PTSD more heavily weight recent prediction errors (as opposed to older prediction errors) when estimating associability values; this may be due to a higher prior belief in the likelihood that prediction errors will change over time.

6) Simulations revealed that associability weight did not affect performance. Is higher associability weight adaptive, maladaptive, or neither?

7) Regarding neural data: Was the associability value that was used as a regressor the updated value for the current trial (i.e. incorporating prediction error on the current trial)? If so, it would reflect the unsigned prediction error on the current trial, especially if the associability weight is high. Is it possible that the neural results simply reflect a stronger response to unsigned prediction error, as opposed to signed prediction error, in PTSD? Even if the updated value was not used, it seems that the associability value should be correlated with unsigned predication error on current trial.

8) Was there a brain-behavior relationship independent of PTSD. It is also not clear whether adding PTSD symptoms to the brain-behavior relationship model improved model fit because the brain-behavior interaction was stronger in PTSD or whether the brain-behavior interaction was weaker in PTSD.

9) In the Introduction, the correspondence between the model and clinical constructs related to PTSD could be further clarified conceptually. Are unexpected events, reminders of negative events, and threats all being equated? Is associability the same as attention to threat?

[Editors' note: further revisions were requested prior to acceptance, as described below.]

Thank you for resubmitting your work entitled "Associability-modulated loss learning is increased in posttraumatic stress disorder" for further consideration at *eLife*. Your revised article has been favorably evaluated by a Senior Editor, Michael Frank as the Reviewing Editor, and two reviewers.

The manuscript has been improved but there are some remaining issues that need to be addressed. Given their specificity and brevity, I've decided to simply append the reviewers’ requests here:

*Reviewer 1:*

1) If the previous version was lacking certain details, this version gave me an impression that they might have overdone it. The authors might want to tighten the Materials and methods (for example, the MCMC part can be placed into supplementary materials).

2) In the response to the reviewers, the authors justified their modeling approach and pointed to previous literatures in terms of including a decay factor in their model. I'm curious whether removing this parameter would significantly changes the results reported in the paper since it seems irrelevant to the scientific question the authors were interested in.

3) In Figure 2, the plotted associability value and prediction error seems to be inversely correlated (albeit these values were generated using a moving average method). But in their model, as the authors stated in their response to the reviewers, current associability was generated using PEs from previous trial. And if they indeed correlate with each other, then how did they look for the neural correlates of PE and associability values at the same time? Also, the y-axis in Figure 2 was labeled as "parameter value", which should be "variable value".

4) I'm confused with Figure 3 and Figure 4, are they both neural correlates of associability values? And the difference is 3D refers to the associability neural correlates independent of PTSD diagnosis but 4B does?

5) The model free logistic regression mentioned that "the outcome history for stimulus A would be (2+1)/5 = 0.6". I have a hard time understanding why it should be defined this way.

*Reviewer 2:*

6) My primary remaining concern is that I am still having difficulty understanding the three-way interaction between neural encoding of associability value, prior trial outcomes, and PTSD symptoms to predict switching behavior (subsection “Neural substrates of associative learning in PTSD and relationship to behavioral choices”, last paragraph). It is difficult for me to understand what it means that the interaction is "positive" in PTSD subjects and "negative" in non-PTSD subjects, and how to interpret this. Since a small loss was coded as 1 and a large loss as 0, does this mean that in PTSD, small losses were associated with a stronger relationship between neural activity and switching, while in non-PTSD, large losses were associated with a stronger relationship between neural activity and switching? If so, how do the authors interpret this? I do not find the figure showing chi-squared values (Figure 5) illuminating on this point.

---

## [Author Response]

Essential revisions:1) Both reviewers noted that the modeling part was potentially more complex than needed. For example, the paper specified reward sensitivity and learning rate. However, these two parameters might covariate in parameter estimation. They also separated the chosen and unchosen option learning rates yet did not run a nested model comparison to test whether these additional parameters are necessary. From an empirical perspective, the authors might want to restrict their model to a simpler one or provide enough evidence to support a more complicated one. Also, Equation 3 does not seem to make sense as currently written.

We appreciate the feedback. The reviewers point out three broad areas for clarification regarding our modeling approach: i) potential covariation of parameter estimates; ii) justification of the final model used; and iii) equation 3. We discuss each of these in turn below.

i) Potential covariation of parameter estimates. We recognize the possibility that model parameters such as reward sensitivity and learning rate may not be uniquely identifiable under some conditions. We tested this potential issue with our data and estimation approach by examining parameter recovery and did not find problems with estimating parameters accurately. That is, the parameters were recovered based on data simulated from a range of parameter values, indicating that each parameter is identifiable (see Figure 3—figure supplement 1 for parameter recovery for learning rate, reward sensitivity, associability weight, decay parameters). We have expanded on the parameter recovery analysis in the text:

“To test whether model parameters were independently recoverable and did not covary in a way that would prevent accurate estimation of each parameter separately, data were simulated based on different values of parameters along the range seen in participants’ data.”

In our experience, the model estimation approach taken in our paper (Hamiltonian Monte Carlo as implemented in Stan) deals with potentially correlated parameters better than other estimation approaches, and avoids the problem of poor identifiability with these parameters. We have plotted an expanded version of the parameter recovery results (Author response image 1, expanded from Figure 3—figure supplement 1) to show that we are able to separably estimate learning rate and reward sensitivity; this figure illustrates the recovered reward sensitivity parameters at different levels of learning rate (LR) and reward sensitivity (RS).

**Author response image 1. respfig1:** Identifiability of reward sensitivity and learning rate. Learning rate (LR) and reward sensitivity (RS) were simulated at three levels (LR:.666,.75, and.825; RS: 1, 1.25, and 1.5), chosen based on the range of actual participants’ values; additional parameters of associability weight and unchosen learning rate were also simulated at three levels, resulting in 81 different combinations of parameters. The average recovered parameter value for each combination of learning rate and reward sensitivity is plotted above, showing that reward sensitivity is recovered well at each level of learning rate.

In contrast, reward sensitivity and inverse temperature are not identifiable when used in the same model, and so we tested which parameter to retain as a free parameter and which to fix, based on improvements in model fit, and found that allowing reward sensitivity to vary as a free parameter improved fit (see next point), and therefore used the model including reward sensitivity and not inverse temperature.

ii) Justification of final model. The reviewers also suggested the need for further evidence to support the final model. In arriving at the final model, we compared model fit for a basic RL model with parameters of learning rate and inverse temperature versus our model (prior to adding the associability weight parameter), which has parameters of learning rate, reward sensitivity, and unchosen learning rate (now termed decay; see response to comment 2 below). This more complex model fit better (using AICc as a fit metric to penalize the additional complexity), with a protected exceedance probability of ~100%. We have added this analysis to the text:

“…we tested the fit of a basic reinforcement learning model, with parameters of learning rate and inverse temperature, versus our model with parameters of learning rate, reward sensitivity, and decay (prior to adding the additional parameter of associability weight) and found that this more complex model fit better, with a protected exceedance probability of 100%.”

iii) Clarification of equation 3. We apologize for the confusion with Equation 3 – there was a typographical error, and it has been corrected to read:

**R**′_A_(**t**) = ρ **R**_A_(t)

Similarly, Equation 2 has been edited:

𝛅(t) = 𝐑′(t) − 𝐐(t)

2) Please add some additional detail regarding the task to the main text. It is difficult to understand the model without understanding the task. Within the task description in the supplemental materials, several aspects of the task remain unclear:

We regret the confusion. These questions are addressed in turn below, and the text has been edited as indicated. We have merged the task description that was previously in the supplemental methods with the relevant section in the main text Materials and methods. Complete task details are now provided into the Probabilistic learning task section of the Materials and methods.

- What was the probability of better/worse outcomes for the less rewarded stimulus?

The probability of better/worse outcomes for the less rewarded stimulus was.25/.75. We have added these details to the text:

“On each trial, participants were presented with two abstract stimuli. One stimulus had a higher (75%) probability of leading to a better monetary outcome and a lower probability (25%) of leading to a worse monetary outcome, while the probabilities for the other stimulus were reversed (i.e., smaller probability of better outcome and larger probability of worse outcome).”

- What was the probability distribution of payoffs within the specified ranges, i.e. within 20-30 or 70-80 (uniform, Gaussian, etc.)?

The payoffs for the better and worse outcomes were calculated at the beginning of each block, fixed for the remainder each block, initialized at 20, 25, or 30 cents, and 50 cents apart. Better/worse outcome pairs were drawn (with replacement) from a uniform distribution from this set, so for each block payoffs were randomly chosen to be +/-20 and +/- 70, +/-25 and +/-75, or +/-30 and +/-80 cents. We have added these details to the text:

“At the beginning of each block, high and low outcome values were randomly chosen with replacement from uniformly distributed outcome pairs {20,70}, {25,75}, or {30,80} and kept consistent within blocks (the outcome displays indicated ‘You Lose [amount]’ for loss blocks, and ‘You Gain [amount]’ for gain blocks).”

- Were outcomes shown for both chosen and unchosen stimuli? If not, how did subjects learn from the unchosen stimulus?

Outcomes were shown only for the chosen stimulus; participants did not see the outcome of the unchosen stimulus. We acknowledge our original use of the term ‘unchosen learning rate’ was confusing in this context, as it implies updating based on counterfactual learning. Therefore, we have changed the nomenclature of this parameter to a ‘decay’ parameter, as the parameter functions to measure the extent to which participants forget the previously learned value of the unchosen stimuli (i.e., how quickly the value of unchosen options decay, as in: Boorman, Behrens, Woolrich, & Rushworth, 2009; Cavanagh, 2015; Collins & Frank, 2016; Niv et al., 2015; and others).

We have reworded the relevant text and changed the citations to similar models to read: “A decay parameter for the unchosen option B (γ; range 0+) (Boorman et al., 2009; Cavanagh, 2015; Collins and Frank, 2016; Niv et al., 2015) was also included.”

Additionally, we have edited other references to the “unchosen" learning rate parameter to now refer to the “decay” parameter.

We have also clarified this aspect of the task description in the text:

“The participant’s choice was framed for a jittered viewing time of 2-4 seconds, after which the outcome (monetary amount gained or lost) was shown for 2 seconds. Participants were not shown the outcome associated with the unchosen stimulus.”

3) The authors might want to clarify the logic behind titrating the length of the task to achieve certain level of better choice selection rates. Different performance in the task itself is a behavioral marker between the control and PTSD groups. In fact, Figure 2 suggests that there is a difference in performance between the groups. Also, given the design of the task, gain and loss blocks alternated. Within the gain block, associability in general shows a declining trend. Thus, it is important to disentangle a non-specific adaptation signal from the associability signal that the paper focuses on.

As the reviewer comment suggests, patient groups often differ in their speed or other behavioral markers of learning. When designing our learning task we wanted to maximize the likelihood that all participants, regardless of psychopathology, would achieve learning. Therefore, rather than have all participants complete blocks of equal numbers of trials, which may either a) extend beyond learning and lead to task disengagement or b) end prematurely while learning is still ongoing, we allowed block length to vary based on performance. An adaptive design thus allowed us to achieve the goal of examining differences in model-based processes associated with successful learning. We have added information about this to the text:

“As our goal was to examine mechanistic processes associated with learning, including in individuals who may differ in learning, we used an adaptive algorithm to ensure that participants successfully learned the contingencies and continued to learn throughout the task.”

We note that our participants with PTSD and no PTSD were in fact comparable in overall learning (measured by proportion of better options chosen, as noted in the last paragraph of the subsection “Participant characteristics and model- behavioral performance”), block length, and number of blocks completed (subsection “Probabilistic learning task”, second paragraph). We also note that in Figure 2, the shaded area represents +/-1 SEM from the mean for each group and that these areas overlap throughout learning, supporting our quantitative findings of lack of overall performance differences in veterans with and without PTSD.

To address the possibility that associability values are confounded with adaptation or time spent on the task, we correlated trial number against ongoing associability value. These values were not correlated, with an average correlation across subjects of -.116. Additionally, we controlled for trial number when assessing the relationship between associability value and reaction time (subsection “Behavioral model-fitting and relationship of model parameters with PTSD”, second paragraph and Figure 3); trial number itself was not related to reaction time while associability value was. These results provide converging evidence that associability and a general adaptation trend are well dissociated during loss learning in our task.

We have added information about this to the text:

“Trial number was included as a covariate in these analyses to account for general adaptation effects; however, trial number and associability were uncorrelated (average correlation across subjects: *r* = -.12).”

As the associability model did not fit gain learning well, we did not investigate associability during the gain blocks.

4) There are significant problems with the conceptual interpretation of the results:-Based on the equations, associability value is expected (unsigned) prediction error:Equation 6 can be rewritten as 𝛋A(t + 1) = 𝛋A(t) +η(|𝛅 t | – 𝛋A(t))Associability weight is the learning rate in learning this associability value (i.e. in learning expected unsigned prediction error). Presumably, this learning rate should be high if the prediction errors themselves are expected to be more likely to change over time.If this interpretation is correct, many descriptions of associability weight in the text are misleading. For example, "behavioral parameter estimates of associability…increased with PTSD" (Abstract), "[Associability weight] indicates the extent to which the magnitude of previous prediction errors is used to update trial-by-trial associability values", "associability weights, reflecting the degree to which stimulus values are modulated or not by associability during loss learning for each participant". Associability weight is actually a learning rate and so reflects the extent to which recent prediction errors (as opposed to older prediction errors) are weighted in updating associability values. This is different from stating associability weight is a measure of associability or that it reflects the extent to which associability is used to make decisions. This distinction is crucially important to understanding the results conceptually.

We acknowledge the reviewers’ comments on the interpretation of our model and fully agree that associability weight acts as a learning rate for learning associability values.

We believe that two areas in which we were unclear formed the basis of this reviewer comment. Specifically, we were i) inconsistent in specifying when we meant associability *value* of the stimulus vs associability *weight* and also ii) imprecise in how we previously described the function of the associability weight. To address these issues, we have first clarified throughout the text in each mention of ‘associability’ whether associability refers to the associability value of the stimulus or the associability updating weight. Second, we have also clarified that associability weight functions as a learning rate with which the associability value is updated (as the reviewers indicate, this reflects the degree to which recent prediction errors are weighted in updating associability values). Specifically, we have revised the passages noted by the reviewers, and other text throughout, to clarify that the associability weight is a learning rate and, as such, functions as an updating parameter:

Abstract: “Neural substrates of associability value and behavioral parameter estimates of associability updating, but not prediction error, increased with PTSD during loss learning. Moreover, the interaction of PTSD severity with neural markers of associability value predicted behavioral choices.”

Introduction: “Furthermore, PTSD severity, particularly symptoms related to hyperarousal (Lissek and van Meurs, 2014), should correlate preferentially both with enhanced associability updating and with increased activity in neural structures encoding associability (i.e., amygdala and insula).”

Results: “In this model, the associability value κ of a chosen stimulus changes on a trial-by-trial basis based on a combination of the magnitude of previous prediction errors and a static associability weight η, a parameter which varies by participant and indicates the extent to which the magnitude of recent prediction errors updates trial-by-trial associability values…”

Results: “To test this possibility, participants’ individually estimated associability weights, reflecting the degree to which associability values are updated based on recent unsigned prediction errors during loss learning for each participant, and unmodulated learning rates (α) were compared between participants with and without a PTSD diagnosis…”

Results: “Corroborating the behavioral findings of greater associability updating in participants with PTSD, associability value showed a significant relationship with PTSD at an FDR-corrected whole brain significance level…”

Discussion: “We posited that the hypersensitivity may derive from either i) heightened solely error-related learning that increases learning directly from unexpected outcomes, or ii) heightened updating of associability values in which disproportionate attention is dynamically allocated in a way that magnifies the effects of prediction errors in future decision-making.”

Discussion: “neither static error-based learning rate nor neural prediction error responses were associated with PTSD, while dynamic attentional modulation (i.e., associability updating) of loss learning signals increased with PTSD at both the behavioral and neural levels.”

Discussion: “While the increased updating of associability values observed here in PTSD during monetary loss learning…”

5) It seems that associability does not play a stronger role in subjects with PTSD. It also does not appear to be the case that PTSD subjects are systematically overestimating associability values. Rather, subjects with PTSD more heavily weight recent prediction errors (as opposed to older prediction errors) when estimating associability values; this may be due to a higher prior belief in the likelihood that prediction errors will change over time.

To clarify, participants with PTSD exhibited greater associability weights than those without PTSD. We fully agree with the interpretation that these increased weights indicate that participants with PTSD more heavily weight recent prediction errors when estimating associability values. We believe that some lack of precision in describing the function of associability weight in our original submission contributed to this reviewer comment. We clarify below:

Specifically, behavioral parameter estimates of associability *weight* (see also point 4 above) are increased in PTSD participants (shown in Figure 4, main text). This increased associability weight indicates, per the reviewer comment, that PTSD participants more heavily weight recent prediction errors when estimating associability values. We have edited our text to clarify this (see changes listed in response to point 4 above).

This increased associability weight may indeed derive from a prior belief in a greater chance of fluctuating prediction errors, but also could arise from differences in processing prediction errors during the task or from other biases in updating associability values. Future work could manipulate and disentangle these and other potential sources of increased associability updating in PTSD. We have added these possibilities to the Discussion:

“The increased updating of associability values in PTSD may reflect a higher prior belief that changes in prediction error will occur, a difference in processing changes in prediction errors, or a combination, which could also serve as treatment targets.”

6) Simulations revealed that associability weight did not affect performance. Is higher associability weight adaptive, maladaptive, or neither?

We appreciate this question. Whether higher associability weight is adaptive or maladaptive likely depends on characteristics of the environment. That is, in situations involving unpredictable shifts in how often contingencies change, higher associability weight would be adaptive, but in more stable situations it would be maladaptive. Our task design likely falls in between these two extremes, and in environments similar to this task, associability weight is neither adaptive nor maladaptive. A mix of context-dependent adaptive and maladaptive effects has been found in other aspects of altered emotional processing and attention in PTSD (Vythilingam et al., 2007; Zinchenko et al., 2017).

We have added this information to the Discussion:

“While the increased updating of associability observed here in PTSD during monetary loss learning should apply generally to learning from other classes of negative stimuli, it will be important in future investigations to explicitly test the role of associability-based learning for PTSD in the context of threatening or trauma-related cues which have been shown to impair fear extinction in the disorder (Milad et al., 2009; Norrholm et al., 2011) as well as to connect our findings to models of other behavioral disruptions in mood/anxiety (e.g., (Mkrtchian, Aylward, Dayan, Roiser, and Robinson, 2017)) and to explore under what conditions the greater associability weight in PTSD is adaptive versus maladaptive (as in (Vythilingam et al., 2007; Zinchenko et al., 2017)).”

7) Regarding neural data: Was the associability value that was used as a regressor the updated value for the current trial (i.e. incorporating prediction error on the current trial)? If so, it would reflect the unsigned prediction error on the current trial, especially if the associability weight is high. Is it possible that the neural results simply reflect a stronger response to unsigned prediction error, as opposed to signed prediction error, in PTSD? Even if the updated value was not used, it seems that the associability value should be correlated with unsigned predication error on current trial.

The associability value used as a regressor in all neural analyses was the value prior to updating on the current trial and so did not include the current trial’s prediction error. We have clarified this in the Materials and methods section:

“Regressors for imaging analyses used the prediction error, associability value, and probability at trial t, prior to updating associability value for the next trial.”

To test the relationship between unsigned prediction error and PTSD, we ran an additional neural analysis with unsigned prediction error as a trial-by-trial regressor. This analysis showed robust relationships between unsigned prediction error and BOLD activation across all subjects, similar to previous studies (Author response image 2), but when regressed against PTSD diagnosis at the group level, showed *no modulation by PTSD*. This signal partially overlaps with, but is distinct from, the associability value-related brain activation shown in Figure 3.

**Author response image 2. respfig2:** Unsigned prediction error signal independent of PTSD. Neural signaling of unsigned

prediction error across PTSD and control participants, FDR corrected at p<.05 (p<.001 cluster defining threshold). As illustrated, participants showed a robust signal for unsigned prediction error, but this signal was not related to PTSD.</Author response image 2 title/legend>

We additionally tested the relationship between behavioral estimates of associability value and unsigned prediction error. Given the relationship in our model between unsigned prediction error and associability weight, we expected these measures to be related but not collinear. Consistent with this expectation, there was a moderate correlation between associability value and unsigned prediction error (average correlation across participants: r =.333). Using the Fisher’s r to z transformation on this correlation, we tested the relationship of this correlation with the associability weight parameter across participants and again found a moderate relationship (r =.25, p =.04); similarly, the relationship between unsigned prediction error and associability value was nearing a significant difference for participants with versus without PTSD (t_66_ = 1.92, p =.06). Therefore, associability value and unsigned prediction error are related, as would be predicted conceptually, and are more strongly related with higher associability weights. However, associability value and unsigned prediction error remain distinctly separate measurements, sharing around 10% of their variance in our sample on average. Thus, the increased associability weight parameter and increased neural reactivity to associability value in participants with PTSD cannot be solely attributed to unsigned prediction error.

8) Was there a brain-behavior relationship independent of PTSD. It is also not clear whether adding PTSD symptoms to the brain-behavior relationship model improved model fit because the brain-behavior interaction was stronger in PTSD or whether the brain-behavior interaction was weaker in PTSD.

As illustrated in main text Figure 3, neural encoding of associability value is present independent of PTSD. Additional analyses analogous to main text Figure 5but conducted within non-PTSD participants verified a robust brain-behavior relationship independent of PTSD (i.e., interaction of outcomes and neural encoding of associability values predicting switching behavior; insula: χ^2^ = 19.0, p <.001; amygdala χ^2^ = 13.6, p =.001).

As previously reported, across all participants neural encoding of associability value interacted with prior trial outcomes and PTSD symptoms to predict switching behavior (main text Figure 5). The brain-behavior relationship (i.e., interaction of outcomes and neural encoding of associability values predicting switching behavior) was in the opposite direction for the PTSD and non-PTSD participants (positive in PTSD and negative in non-PTSD). These data thus indicate opposite brain-behavior interactions in participants with and without PTSD (rather than a weaker or stronger relationship in one group) whose direction is dependent on diagnosis.

We have added information about this to the Results section:

“The brain-behavior relationship (i.e., interaction of outcomes and neural encoding of associability values predicting switching behavior) was in the opposite direction for the PTSD and non-PTSD participants (positive in PTSD and negative in non-PTSD).”

9) In the Introduction, the correspondence between the model and clinical constructs related to PTSD could be further clarified conceptually. Are unexpected events, reminders of negative events, and threats all being equated? Is associability the same as attention to threat?

We appreciate this point. We did not intend to equate or differentiate among the listed clinical constructs and have attempted to clarify both this and the connections between the model and clinical constructs in the Introduction.

Clinically, the consequences of unexpected negative events and reminders of negative events (particularly trauma) often overlap in individuals with PTSD (Dunsmoor and Paz, 2015), and patients view unexpected, uncontrollable, or trauma-related experiences as threatening because of the connection to previous traumatic events (Craske, Treanor, Conway, Zbozinek, and Vervliet, 2014; Schnurr et al., 2007). We posit that altered associability-based learning in PTSD likely contributes to some of the consequences (particularly hyperarousal symptoms) of threatening events and may lend precision to understanding which of these constructs are most relevant, but is *not* the same as the clinical constructs themselves (including attention to threat). Rather associability-based learning and related model-based approaches provide mechanistic insight as to how these unexpected events, reminders of negative events etc. trigger symptoms and whether they and their sequelae are (or are not) related.

For example, neural and behavioral responses to prediction errors may track well with clinical responses to unexpected events, and responses to associability values of stimuli may track well with responses to reminders of unexpected negative events, and both may be threatening. Although such potential connections are intriguing, we are reluctant (given the sparseness of the clinical literature and the early stages of the model-based evidence) to make strong statements about precise correspondences between model components and such clinical constructs. The data we present here are nonetheless a promising step in this direction and indicate that heightened updating of associability values of stimuli (which cue or remind of previous events) specifically is related to hyperarousal symptoms in PTSD. Future studies will be required to need to empirically and thoroughly test the mapping of connections between model and clinical constructs related to PTSD. We have clarified the relevant portion of the Introduction:

“In particular, clinical and empirical observations have documented the negative impact of salient cues on neural and behavioral functioning in PTSD, including heightened orienting to unexpected events, impaired extinction of learned fear, and unstable attention biases toward perceived threatening stimuli (Aupperle, Melrose, Stein, and Paulus, 2012; Bar-Haim, Lamy, Pergamin, Bakermans-Kranenburg, and van IJzendoorn, 2007; Blair et al., 2013; Morey et al., 2009; Naim et al., 2015). Together, these behavioral alterations in response to unexpected stimuli, uncontrollable reminders of trauma, and other negative, threatening, and trauma-related events point to PTSD as a disorder of disrupted learning from reminders of negative events; however, the specific components of anomalous learning in PTSD remain unknown.”

And:

“Computational model-based approaches to learning provide a mechanistic framework for understanding the detrimental impact of unexpected negative stimuli and reminders of negative events in PTSD.”

We have also incorporated possible extensions of our model to clinical concepts in the Discussion:

“The clinical manifestation of hyperarousal in PTSD includes excessive sustained vigilance (American Psychiatric Association, 2000) and resembles what heightened updating of associability values would predict; specifically, reminders of trauma can be conceptualized as stimuli associated with unexpected negative outcomes, with the greater associability updating seen here in PTSD causing stimuli associated with a recent history of these surprising outcomes to command greater attention and increased updating.”

[Editors' note: further revisions were requested prior to acceptance, as described below.]

The manuscript has been improved but there are some remaining issues that need to be addressed. Given their specificity and brevity, I've decided to simply append the reviewers’ requests here:Reviewer 1:1) If the previous version was lacking certain details, this version gave me an impression that they might have overdone it. The authors might want to tighten the Materials and methods (for example, the MCMC part can be placed into supplementary materials).

We appreciate the concern for the need to balance brevity and explanation in the Materials and methods section. In preparing the resubmission, we reformatted portions of the manuscript to conform with *eLife* formatting guidelines. We acknowledge that combining the main text methods and supplementary methods as a part of this reformatting has made the methods somewhat unwieldly, and so have moved some of the methods, specifically those pertaining to MCMC parameter estimation and model free behavioral analyses, to Appendix 1: supplementary methods.

2) In the response to the reviewers, the authors justified their modeling approach and pointed to previous literatures in terms of including a decay factor in their model. I'm curious whether removing this parameter would significantly changes the results reported in the paper since it seems irrelevant to the scientific question the authors were interested in.

Thanks for this comment. We assert that the decay parameter is relevant to our question and explains significant aspects of behavior in our task, as evidenced by the improvement in model fit. Consistent with the role of this parameter in governing the retention of unchosen values, the decay parameter is highly correlated with participants’ model free proportion of switching between options (r =.664, p <.001), indicating that this parameter is related to important aspect of participants’ behavior that should not be neglected in our model. However, as we did not hypothesize that the retention of unchosen values would be responsible for different learning patterns in PTSD, we did not focus on this parameter when examining learning differences in PTSD.

We nonetheless tested the robustness of the results without the decay parameter (and preserving associability weight and learning rate), as suggested by this reviewer. To do so, we used the poorer fitting model without the decay parameter, and examined the relationship between the individual associability weight parameters estimated from this model with the associability weight parameters from the full model. The associability weight parameters from these two models were highly correlated, r =.75, p <.001, suggesting that the inclusion of the decay parameter did not have a large effect on estimates of the associability weight parameter. We then tested the relationships between PTSD and loss associability weight and learning rate, as described in the primary behavioral analyses, with parameters from the model without the decay parameter. The relationship between PTSD and associability weight remained significant (t_62_ = 3.36, p =.001), and the relationship between PTSD and learning rate remained nonsignificant (t_62_ =.035, p =.7). Therefore, the inclusion of the decay parameter does not substantively affect the results or conclusions reported in the paper.

3) In Figure 2, the plotted associability value and prediction error seems to be inversely correlated (albeit these values were generated using a moving average method). But in their model, as the authors stated in their response to the reviewers, current associability was generated using PEs from previous trial. And if they indeed correlate with each other, then how did they look for the neural correlates of PE and associability values at the same time? Also, the y-axis in Figure 2 was labeled as "parameter value", which should be "variable value".

We fully agree with the importance of ensuring that associability value and prediction error measure separable influences on behavior. As this reviewer notes, Figure 2 uses a moving average method and so does not reflect the exact trial-wise relationship between prediction error and associability value. The actual relationship between signed prediction error and associability value is similar to the relationship between unsigned prediction error and associability value that was discussed in the previous response to reviewers- it is present but minor. Specifically, the average correlation between associability value and signed prediction error across participants is -.336, meaning that these two variables share about 11% of their variance. The independence of these measures is reflected in the fact that adding prediction error as the first parametric modulator, and associability value as the second parametric modulator, to first level imaging analyses results in a similar relationship at the group level between associability value and PTSD as when prediction error is not included.

We appreciate the feedback on the figure axis label – Figure 2’s y-axis now reads ‘variable value’ as suggested.

4) I'm confused with Figure 3 and Figure 4, are they both neural correlates of associability values? And the difference is 3D refers to the associability neural correlates independent of PTSD diagnosis but 4B does?

The reviewer is correct. Both are correlates of associability value, and Figure 3 shows the intercept term in the regression of PTSD on neural associability values, and so shows the neural correlates of associability value independent of PTSD diagnosis. Meanwhile, Figure 4 shows the PTSD term in this regression and so shows the relationship between PTSD diagnosis and neural associability values.

We have clarified this distinction in the description of these results:

“…neural encoding of associability values showed a significant relationship with PTSD at an FDR-corrected whole brain significance level in a network of regions including bilateral amygdala and insula, hypothesized areas of relevance for PTSD and associability-based learning^1,12^ (Figure 4; Supplementary file 1, Table 1C; see Figure 3 for associability-related signaling across participants after accounting for PTSD and covariates).”

and in the legend for Figure 3:

“(D): Associability signaling independent of PTSD and covariates, displayed at p <.05 FDR corrected.”

5) The model free logistic regression mentioned that "the outcome history for stimulus A would be (2+1)/5 = 0.6". I have a hard time understanding why it should be defined this way.

As part of model free analog of associability modulated learning, we calculated expectedness as the probability of receiving the better outcome with a particular stimulus choice, and then used the difference between this expected outcome and whether or not the outcome was actually better. The outcome history for a particular stimulus in this analysis was therefore intended as the probability that choosing a stimulus would lead to the better outcome the next time it was chosen and was defined the sum of a) better outcomes after choosing that stimulus (2 in the quoted example) and b) worse outcomes after choosing the other stimulus (1 in the example), divided by c) the number of choices to that point in the block (5 in the example). This value approximated the predicted outcome on the next trial that this stimulus was chosen, and was subtracted from the actual outcome to represent the level of surprise or unexpectedness for use in the logistic regression.

Reviewer 2:6) My primary remaining concern is that I am still having difficulty understanding the three-way interaction between neural encoding of associability value, prior trial outcomes, and PTSD symptoms to predict switching behavior (subsection “Neural substrates of associative learning in PTSD and relationship to behavioral choices”, last paragraph). It is difficult for me to understand what it means that the interaction is "positive" in PTSD subjects and "negative" in non-PTSD subjects, and how to interpret this. Since a small loss was coded as 1 and a large loss as 0, does this mean that in PTSD, small losses were associated with a stronger relationship between neural activity and switching, while in non-PTSD, large losses were associated with a stronger relationship between neural activity and switching? If so, how do the authors interpret this? I do not find the figure showing chi-squared values (Figure 5) illuminating on this point.

We apologize for the confusion. As previously reported, across all participants, neural encoding of associability value interacted with prior trial outcomes and PTSD symptoms to predict switching behavior (main text Figure 5). To parse this interaction, we examined the effects of the interaction of neural associability value with PTSD diagnosis outside of the interaction with previous outcomes in predicting switching behavior. The main effect in this model of the effect of neural associability value in predicting switching was negative for both ROIs (insula: z = -2.58; amygdala: z = -2.02), meaning that greater neural reactivity to associability value in these ROIs predicts less overall switching independent of previous outcomes. However, the interaction with PTSD for these ROIs was positive (z values of 2.39 for insula and 1.96 for amygdala), indicating that greater neural reactivity to associability value is related to a greater tendency to switch choices in people with PTSD, again independent of previous outcomes. Therefore, participants with PTSD show an overall higher tendency to switch with higher neural responsivity to associability values relative to controls.

The remaining portion of the interaction, the effect of the value of previous outcomes by diagnosis, depends on the region (amygdala versus insula), which was unclear in our previous response. For amygdala responsivity to associability value, participants with PTSD had a greater relationship between neural responsivity and switching behavior following large losses compared to small losses, as did the controls (see Figure 5—figure supplement 2). For insula responsivity to associability value, participants with PTSD had a smaller relationship between neural responsivity and switching following large versus small losses (Figure 5—figure supplement 2). Our previous response primarily examined the relationship between insula responsivity and switching, and so we apologize for the imprecision in our response. The difference in this part of the interaction between these two brain regions is interesting and could be followed up in future work.

We have added information describing the direction of the interaction to the manuscript:

“A secondary analysis inspecting the components of this interaction revealed overall greater tendency to switch with greater associability value-related neural activation in participants with PTSD relative to controls, with additional effects of large relative to small outcomes leading to a greater tendency to switch in controls in the amygdala and insula ROIs and in PTSD in the amygdala ROI; the effect of outcomes in PTSD in the insula ROI was reversed (Figure 5—figure supplement 2).”

We have also added a plot of the effects of neural responsivity to associability value on switching for each level of outcomes (small vs. large loss) and group (PTSD vs. control) as Figure 5—figure supplement 2.

We agree with this reviewer that understanding the interaction effects is important; however, the primary finding from this analysis was that neural responsivity to associability value, and not prediction error, interacted with hyperarousal and avoidance/numbing, but not reexperiencing, symptoms of PTSD to predict behavior. Due to the complexity of the three-way interaction, the multiple comparisons presented, and concerns about unreliability of specific parameter estimates in multilevel models, we believe the chi-squared values are a straightforward presentation of the primary aspect of the interaction results. For readers interested in a further explanation of the interaction, though, we do agree that additional explication is useful and so have added the text and figures referenced above as supplements to Figure 5.